



# Shortening of the Arctic cold air outbreak season detected by a phenomenological machine learning approach

Filip Severin von der Lippe[1], Tim Carlsen[1], Trude Storelvmo[1], and Robert Oscar David[1]

[1]University of Oslo

**Correspondence:** Filip Severin von der Lippe (fslippe@uio.no)

**Abstract.** Marine cold air outbreaks (CAOs) frequently occur in the Arctic when cold air moves over the relatively warm ocean, resulting in large turbulent fluxes, instability and cloud formation. Given the high frequency of CAOs during the Arctic winter, the associated clouds have a large impact on the region's radiative balance. Due to Arctic warming, the prevalence of CAOs and their clouds may change, impacting the Arctic radiative balance and potentially amplifying or mitigating local and global warming.

To better understand how CAO clouds respond to Arctic warming, this study has developed a phenomenological CAO cloud classification tool that utilizes machine learning methods to identify closed and open cell clouds in CAOs from MODIS satellite imagery. This new approach achieves better performance in identifying CAO clouds compared to the marine cold air outbreak index calculated using MERRA-2 reanalysis, with accuracies of 85.4 % and 78.0 %, respectively. The new approach has revealed frequent CAO cloud formation in regions of high sea surface temperatures, with occurrence maxima along the Norwegian coast and the Northern Atlantic region south of Iceland. Furthermore, the approach reveals trends in CAO cloud cover that suggest a shortening of the CAO season, characterized by an approximate 10 %, increase in cloud coverage during winter and a nearly 20 % decrease during the shoulder months over the past 25 years. These trends suggest a positive radiative feedback during winter in response to climate change, underscoring the importance of further investigating these clouds to understand the trajectory of future Arctic climate.

## 1 Introduction

Clouds in polar regions are often associated with marine cold air outbreaks (CAOs). These clouds form in the marine boundary layer (MBL) when cold and dry air from snow- and ice-covered regions moves over the relatively warm ocean. This produces a turbulent environment where large latent and sensible heat fluxes lead to the formation of clouds. Near the sea ice edge, these clouds form long cloud streets of densely packed closed cell stratocumulus, which transition into open cell broken cumulus when they traverse the open ocean (Brümmer, 1999; Geerts et al., 2022). These CAO clouds have a large impact on the surface radiative energy balance through their extensive coverage (Fletcher et al., 2016b). This is observed through their high albedo compared to the underlying dark ocean, reflecting incoming solar radiation back to space (cooling effect), and absorption and re-emission of outgoing terrestrial radiation (warming effect).





As the Arctic has experienced significant warming in recent decades (Serreze and Barry, 2011), it is crucial to study how CAOs may be affected. Specifically, the strength of CAOs is projected to change (Landgren et al., 2019), which could affect cloud properties (Murray-Watson et al., 2023) and influence future warming. Changes in CAO clouds could drastically impact the Arctic radiative balance, potentially amplifying or damping the strong warming observed in this region.

To better understand the development of CAO clouds, they have been extensively studied through modeling, in situ observations and satellite studies (e.g., Hartmann et al., 1997; Abel et al., 2017; Geerts et al., 2022; Wu and Ovchinnikov, 2022). Especially, the transition from dense closed-cell stratocumulus to open-cell broken cumulus has been investigated (e.g., Abel et al., 2017; Yamaguchi et al., 2017; Tornow et al., 2021). As the denser closed cells have a higher albedo than the open cells (McCoy et al., 2017), the processes influencing the break-up to open cells become of high importance when studying the radiative impact of CAO clouds.

Several studies have suggested the onset of precipitation as a main driver for cloud break-up (e.g., Abel et al., 2017; Yamaguchi et al., 2017; Tornow et al., 2021), while others suggest that changes in the MBL stability (McCoy et al., 2017), such as from increasing sea surface temperatures (SSTs), drive the break-up. What these processes have in common is the drying and dissipation of the closed cell cloud layer. This may occur through decoupling of the cloud layer from the moisture-supplying surface (Bretherton and Wyant, 1997), through water loss due to precipitation which may also lead to decoupling (Abel et al., 2017), or by favoring the formation of cumuliform clouds through the moistening and destabilization of the lower MBL (Stevens et al., 1998). While Abel et al. (2017) focused on a single CAO, introducing uncertainties regarding its universality, others have analyzed multiple CAOs by utilizing the marine cold air outbreak index $M$ (Kolstad and Bracegirdle, 2008). This index measures the instability of the MBL and is calculated as the difference between the surface potential skin temperature and the potential temperature of a chosen pressure level, typically 850 hPa (i.e. Papritz and Spengler, 2017). By utilizing reanalysis products such as Modern-Era Retrospective analysis for Research and Applications, Version 2 (MERRA-2, Gelaro et al., 2017), which provide global climate and weather data of the past at up to hourly resolution, earlier studies have defined a CAO as a model grid point where the index $M$ is positive, indicating instability and the possibility for clouds (e.g. Fletcher et al., 2016b; Murray-Watson et al., 2023). This method provides an easy way to find the location of CAOs to use for further analysis, such as investigating cloud break-up. Despite the ease of use, reanalysis data introduces model biases especially in remote regions such as the Arctic with limited available observational data. In addition, uncertainties arise from the fact that a positive index does not necessarily result in the existence of a CAO cloud. This motivates the introduction of a phenomenological approach to defining CAOs that is based on the existence of clouds and that is free of the biases introduced by modeling and reanalysis.

Clouds within CAOs typically cover large areas and are easily distinguishable from other clouds due to their cellular structure. This makes them easy to spot from satellite images (e.g. Fig. 1b), which in turn can be used to better understand their coverage and radiative impact. For the Arctic, this requires a polar-orbiting satellite such as Terra which since 24 February 2000 has provided multiple products of the surface, atmosphere and clouds through the onboard Moderate Resolution Imaging Spectroradiometer (MODIS) instrument. This instrument provides near-daily surface, sea ice, ocean and atmosphere data products of the entire Earth (King et al., 1992), and has been extensively used through its measurements of 36 radiance bands




in the solar and thermal infrared spectral range. The 36 MODIS radiance bands are calibrated and provided as 5 minute swaths
with a wide viewing angle of 55°, giving a total coverage of 2330x2030 km. This extensive coverage makes MODIS an optimal
instrument for classification of CAO clouds.

Utilizing MODIS products, a human can hand-label CAO clouds of interest, rather than using meteorological parameters
to predict the possibility of a cloud. Such a labeling approach would be time-consuming and could introduce significant
subjective bias from the labeler, as demonstrated in other cloud classification tasks (Stevens et al., 2020). To automate the
cloud classification process, machine learning methods may be utilized. Wood and Hartmann (2006) introduced a supervised
neural network (NN) utilizing the MODIS liquid water path (LWP) retrievals to classify closed and open cells in the subtropics.
Despite its application in the study of CAOs (McCoy et al., 2017), this NN requires MODIS daytime retrievals, leading to a
lack of data during the polar night. As Arctic CAOs are most frequent during the dark winter season from late autumn to early
spring (Fletcher et al., 2016a), this model remains inapplicable for Arctic CAO studies.

Utilizing daytime MODIS calibrated radiances, Kurihana et al. (2022) developed an unsupervised machine learning ap-
proach comprising an autoencoder (Hinton and Zemel, 1993) and hierarchical agglomerative clustering for cloud classification.
Although this study relied on certain radiance bands that are only available during daytime, the method can be adapted to elimi-
nate this dependency. By modifying the approach of Kurihana et al. (2022), it becomes possible to phenomenologically classify
wintertime CAO clouds using nighttime available bands in the thermal infrared. Specifically, band 31 (10.780 - 11.280 $\mu$m)
in the thermal infrared can be selected as it is widely used for MODIS cloud classification algorithms given its sensitivity to
clouds (Frey et al., 2008). As this approach is unsupervised, it also reduces the requirement of human labeling for training,
reducing subjectivity and possible restrictions from the training dataset in producing accurate classifications. By developing a
similar tool to Kurihana et al. (2022), this study will not only help in understanding drivers of cloud break-up within CAOs,
but also provide an accurate database of CAO clouds for other studies, such as investigating past changes in CAO cloud cover
to provide future projections of the radiative impact of CAOs.

This study aims to introduce a new phenomenological CAO cloud classification tool called CAOnet, utilizing a NN and 25
years of MODIS satellite imagery. CAOnet will provide a valuable CAO climatology database for future studies of CAO clouds,
mitigating uncertainties introduced by traditional reanalysis methods. Along with a $M$ index optimized for cloud detection,
CAOnet will be employed to explore the potential radiative impact of changes in CAO cloud cover in response to projected
climate change.

## 2 Methods

### 2.1 Data and model development

#### 2.1.1 MODIS data preparation

Swaths of MODIS calibrated radiances (band 31, 10.780 - 11.280 $\mu$m) in the thermal infrared from the satellite Terra were
utilized to provide a database for cloud classification during an extended winter period from September to May.



Each MODIS swath has a resolution of 1354x2030 pixels, that translates to 1 km resolution at nadir, decreasing to 4.8 km at the scan extremes. To keep the resolution as close to 1 km as possible for all pixels, the swath width was decreased from 1354 to 1024 pixels. This resulted in a resolution of approximately 2.05 km at the scan extremes, balancing uniform pixel res-

95 olution while keeping daily swath coverage in the Arctic north of 55 °N. Although additional MODIS swaths from the satellite Aqua could have been utilized to further enhance pixel uniformity, this was avoided to minimize storage and computational requirements. Finally, up to four temporal subsequent swaths were combined to extend the swath size from five to 20 minutes, resulting in a resolution of up to 1024x8120 pixels.

In order to make classifications of smaller regions containing closed and open cells, the combined satellite swaths were

100 split into smaller image patches of 128x128 pixels each. Such a patch was large enough to cover multiple closed or open cloud cells, which made it possible for a human and the classification model to distinguish cellular structures from other cloud fields. By utilizing multiple swaths acquired between 1 March 2000 and February 28th 2025 split into patches, a database for a classification model was created.

### 2.1.2 Developing a phenomenological classification model

To classify the satellite images and find CAO clouds, an unsupervised machine learning approach based on the work of Kurihana et al. (2022) was developed. While Kurihana et al. (2022) used hierarchical agglomerative clustering, this study utilizes K-means clustering for faster computation and the ability to train a model for predicting unseen data. K-means is an unsupervised machine learning algorithm that classifies data into a user-defined number of clusters. It operates by optimizing cluster centroids during training, and assigning a given input to the cluster with the nearest mean. In terms of the satellite image

patches, this mean equals the mean pixel value of each patch. For the case of looking at clouds, the mean pixel value may not efficiently describe cloud structure or type, giving meaningless classifications. By giving K-means more input than just the original image, it may create more meaningful clusters, especially if the input describes the features that are most important for the specific image. Such a feature could describe image contrast, cloud cell size, or brightness, as well as features that are meaningless to humans but still helps informing correct classification.

Similar to Kurihana et al. (2022), an autoencoder was used to extract dimensionally reduced information incorporating the most important information of the input patches. This is a convolutional neural network that comprises two main components (see Fig. 1a). The first component is the encoder, which performs compression on a given input image, producing a compressed feature representation. This is fed to the decoder, whose goal is to decompress the encoded features into an output that resembles the original input image. During training, these two components work together to most accurately reproduce the original input

image. The encoder achieves this through saving the most significant information in the compressed feature representation, helping the decoder to produce an accurate image reconstruction. The compressed feature representation produced by the encoder was employed in further classification tasks, aiding K-means in making meaningful classifications. Thus, the encoder and K-means clustering comprised the two main components of the classification model called CAOnet, as visualized in Fig. 1b.

A simplified structure of the autoencoder used in the subsequent analysis is shown in Fig. 1a. The encoder performs four

dimensionality reductions, with each reduction halving the width and height of the input to that layer. Simultaneously, the



number of features are increased from the single-feature band 31 input patch, to 16, 32, 64 and finally 128 features. While some of the final 128 features could be understandable to a human, others are likely to be meaningless. Nevertheless, this number was found to aid K-means in creating the most meaningful clusters. For each dimensionality reduction, the input passes through residual blocks, which have been shown to mitigate performance degradation in deep neural networks (He et al., 2015). Within these residual blocks, three convolutions and batch normalization are performed before the output $x$ is processed through a leaky rectified linear unit activation function:

$$f(x) = max(0.3x, x).$$

This activation function introduces nonlinearities to the NN, enhancing its ability to understand complex patterns.

After passing through the encoder, an input patch of 128x128 pixels results in 128 features of 8x8 pixels. Utilizing this encoded output, the decoder performs transposed convolutions and upsampling. This results in an autoencoder with 16 trainable convolutional layers, with four of these located in the decoder. In total, the autoencoder comprised 635,953 trainable parameters, which were optimized for image reconstruction and precise compressed feature representations. This helped K-means to make meaningful cloud classifications, assigning a single label to each 128x128 image patch.

### 2.1.3 Training the classification model

To train CAOnet comprising an autoencoder and K-means clustering, a total of 15,200 swaths, split into 600,000 patches was used. This data covered the period from November to April for the years 2018 to 2023. First, a randomly sampled 85/15 split of the data was performed to create a training and test dataset. The autoencoder was individually trained on the whole training subset, and evaluated using the test split after one pass through the entire training dataset. The loss function optimized during training was a combination of mean squared error and Sobel loss similar to that used by Kurihana et al. (2022). Training of the autoencoder was stopped once the test loss had converged.

The encoder of the trained autoencoder was used to produce compressed feature representations that assisted K-means during training in producing meaningful clusters. In total, ten K-means models were trained, each with different numbers of predefined clusters ranging from 7 to 16. Finally, the classification of the ten K-means models were evaluated against a hand-labeled dataset to determine which of their clusters most closely aligned with CAO clouds, as further explained in the next section. Figure 1b shows an example of the satellite image classification process using a trained K-means model with 7 predefined clusters. Here, the red and pink clusters were evaluated to most closely align with CAO clouds.

### 2.1.4 Evaluating the classification model

Even though the autoencoder and K-means clustering are unsupervised machine learning methods, their classifications require inspection and evaluation to acquire meaningful information about CAO clouds. To evaluate K-means clustering with different predefined numbers of clusters, a hand-labeled dataset was generated. To make sure there were no changes in the model's ability to classify CAO clouds as a result of potential climatological shifts, five datasets covering 5-year periods from 2000 to 2024 were created. To guarantee that each of the five subsets contained cases with and without CAOs, one K-means model



**(a)**

**(b)**

**Figure 1.** A visualization of a typical CAO and how it is processed by the unsupervised classification model. In panel (a) an image patch is fed through the autoencoder which before training produces noise (orange). The trained autoencoder has optimized its trainable parameters leading to an accurate reproduced image patch (purple). In panel (b), a classification example is shown, utilizing the encoder of the trained autoencoder. Two patches (marked in red and blue) are followed from extraction to encoding, K-means clustering and a final classified image. The blue patch containing typical CAO cellular cloud formation is classified as red, while the high clouds in the red patch are classified as brown. In this example, K-means clustering with seven clusters was used, with the red and pink cluster evaluated to most closely align with CAO clouds. Black regions of the classified image denote land, sea ice or regions outside of the study domain.





**Table 1.** Table showing the origin of the swaths making up the evaluation dataset.

| Evaluation data subset | swaths per 5-year period | total swaths |
|---|---|---|
| CAO | 25 | 100 |
| No CAO | 25 | 100 |
| Random | 25 | 100 |
| Mixed | 25 | 100 |
| **All** | **100** | **500** |

with 14 clusters was randomly selected to identify satellite swaths containing CAOs. This involved manually inspecting six example images of CAOs, revealing four clusters from the example model aligning with CAO clouds (see Fig. C1).

By utilizing the randomly chosen K-means model with 14 predefined clusters, images containing CAOs (more than 30 coherent CAO patches) and those without CAOs (fewer than 2 coherent CAO patches) were identified based on predictions of the four clusters aligning with CAOs. This process resulted in CAO subsets and no-CAO subsets for each 5-year period. Additionally, random subsets were created, including randomly sampled swaths from each 5-year period. The final mixed subsets contained randomly sampled swaths for each 5-year period with equal probability corresponding to the three other

subsets. In total, this resulted in 500 swaths, with 100 swaths for each 5-year period. From these 100 swaths, 25 swaths corresponded to each of the CAO, no CAO, random, and mixed subsets, as shown in Tab. 1.

The created evaluation dataset was presented to a labeler through an interactive website. The instructions were to draw regions where they believed they observed CAO-related clouds such as closed and open cells. The evaluation results were then compared with the different K-means models to assess which of their clusters most accurately represented CAOs by calculating

several score metrics.

### 2.1.5 Evaluation score metrics

The score metrics used to choose the most optimal clusters for CAO cloud classifications were the Matthews correlation coefficient (MCC), precision, recall and $F_\beta$ scores. The MCC is a measure of association of two binary variables and defined through the number of true positives (TP), false positives (FP), true negatives (TN) and false negatives (FN) as:

$$\text{MCC} = \frac{\text{TP} \times \text{TN} - \text{FP} \times \text{FN}}{\sqrt{(\text{TP} + \text{FP})(\text{TP} + \text{FN})(\text{TN} + \text{FP})(\text{TN} + \text{FN})}}. \tag{1}$$

Precision is defined as the fraction of relevant predicted instances over all retrieved instances, while recall is defined as the fraction of relevant predicted instances over all relevant instances:

$$\text{precision} = \frac{\text{TP}}{\text{TP} + \text{FP}} \tag{2}$$

$$\text{recall} = \frac{\text{TP}}{\text{TP} + \text{FN}}. \tag{3}$$





Utilizing the precision and recall, the $F_\beta$ score is a measure of predictive performance in binary classification analysis and calculated as:

$$F_\beta = (1 + \beta^2)\frac{\text{precision} \times \text{recall}}{\beta^2 \times \text{precision} + \text{recall}}. \tag{4}$$

Here, $\beta$ is a parameter which denotes the relative importance of recall compared to precision. Although the $F_1$ score with $\beta = 1$ is typically used, a final $\beta$ of 1/1.75, valuing precision 1.75 times more than recall was chosen. This reflects a willingness to miss some true positives in order to reduce the number of false positives. It is expected that the model will be able to accurately classify clear CAO cases, and by prioritizing precision over recall, the risk of false positives biasing the data is minimized even if it results in oversight of less typical CAO cloud structures.

Finally, a combination of the $F_\beta$ and MCC score can be calculated for a final evaluation score. By normalizing $F_\beta$ and $MCC$ over all models to be evaluated, the MCC and $F_\beta$ scores equally contribute to the combined score ($s$) given as:

$$s = \frac{\text{MCC}/\text{MCC}_{\text{max}} + F_\beta/F_{\beta,\text{max}}}{2}, \tag{5}$$

where $\text{MCC}_{\text{max}}$ and $F_{\beta,\text{max}}$ are the highest MCC and $F_\beta$ scores found for all the evaluated models.

### 2.1.6 Producing a binary CAO classification database

After the best performing K-means model and associated CAO clusters were chosen, the final optimized CAOnet was settled on to make predictions on all MODIS swaths for autumn (Sep., Oct., Nov.), winter (Dec., Jan., Feb.) and spring (Mar., Apr., May) between March 2000 and February 2025. These predictions were re-gridded onto a 100 km resolution grid of the Arctic, creating a daily binary CAO classification database.

Potential biases may be introduced to this database through uneven MODIS coverage, confusion between clouds, land, and sea ice in coastal regions, and the uneven distribution of missing data. To address these issues, several processing methods were implemented.

First, to prevent uneven MODIS coverage, a day was defined as extending from 01:00 UTC to 00:59 UTC the following day. This ensured that all grid points in the study areas experienced at least one overpass each day, given the orbit of the Terra satellite.

Second, to avoid bias from more overlapping MODIS swaths at higher latitudes, as shown in Fig. 2b, random sampling among overlapping swaths was performed. This guaranteed that every grid point received only one classification per day, resulting in a uniform database.

Third, to minimize the influence of sea ice and land on cloud classifications, patches were discarded if their open ocean fraction was less than 95%. To calculate this fraction, the sea ice concentration dataset from Nimbus-7 SMMR and DMSP SSM/I-SSMIS Passive Microwave Data, Version 2 (DiGirolamo et al., 2022) was used. This resulted in lower data coverage for grid points near the sea ice edge and land, as shown in Fig. 2a.



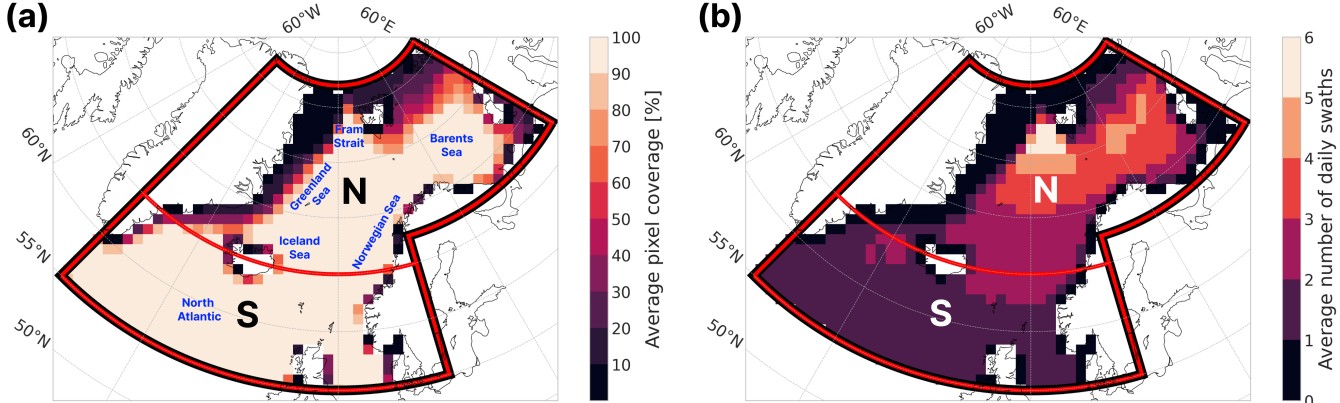

**Figure 2.** Average daily grid point coverage (a) and number of MODIS swaths (b). Low coverage can be found in coastal areas where patches typically contain less than 95% open ocean. S and N define a southern and northern subregion for subsequent analysis. Names of the focus regions for this study are marked in blue.

Finally, to prevent the influence of missing data and the variability of the sea ice edge from affecting further analysis, all grid points with more than 10% missing data over the whole study period were discarded. This led to the final grid points shown in the 90-100% bin in Fig. 2a.

### 2.1.7 Evaluating CAO classification based on $M$ index

To compare the phenomenological approach to a reanalysis approach for CAO cloud classification, CAO classifications were made using the $M$ index calculated from MERRA-2 reanalysis. To optimize its performance, various index thresholds ($M_{\mathrm{thr}}$) were evaluated against the evaluation dataset:

$$M = \theta_{SST} - \theta_{850} \geq M_{\mathrm{thr}}, \tag{6}$$

where $\theta_{SST}$ represents the potential sea surface temperature and $\theta_{850}$ the potential air temperature at 850 hPa. The threshold yielding the best score ($s$) according to Eq. 5 was then used to create a $M$ index binary CAO classification database for the same grid points as those used in CAOnet. Here, the average $M$ index for any given day and grid point was used. To ensure a fair comparison between the two CAO databases, only days and regions with MODIS coverage were considered for the $M$ index as well.

### 2.2 CAO climatology

To produce a climatology of CAO clouds, both CAOnet and the MERRA-2 $M$ index were used. Similar to Papritz and Spengler (2017), an extended winter stretching from November through April was used for the climatological analysis. For each grid point, a relative frequency of occurrence (RFO) was calculated, representing the fraction of days with CAO coverage relative to the number of days MODIS had coverage for that grid point. For MERRA-2, two $M$ index thresholds of 0 K and 3.75 K



called $M_0$ and $M_{3.75}$ were used, where the latter threshold was selected based on the model evaluation as described in Sect. 2.1.7.

## 2.3 Trend analysis

To include the study of latitudinal variations of cloud cover trends and their impact on the radiative balance, the entire domain was divided into a southern and northern region, as shown in Fig. 2. The northern region was chosen based on typical CAO trajectories, which indicate strong CAOs and high RFO (Murray-Watson et al., 2023). This region is also situated closer to the sea ice, increasing the possibilities for larger concentrations of closed cells. In contrast, the southern region was chosen for its weaker CAOs (Papritz and Spengler, 2017) and its distance from the sea ice edge, suggesting higher concentrations of open cells. Finally, the binary classification databases were used to make three daily CAO cloud coverage datasets: the entire region, the southern region and the northern region.

Utilizing the CAO cloud cover fraction of each region, trends were calculated using the Theil-Sen trend estimator $T_{TS}$ (Theil, 1950; Sen, 1968). This trend estimator is non-parametric, meaning it is independent of the distribution of the data, making it widely applied in climate data analysis (Gilbert, 1987; Yue et al., 2002; Collaud Coen et al., 2020). It is estimated using daily mean coverage of CAO clouds across the three regions, calculated for all pairs of sample points as:

$$T_{TS} = \text{median} \left( \frac{y_i - y_j}{x_i - x_j} \right), \tag{7}$$

where $y_i$ denotes the cloud cover fraction on day $x_i$. Additionally, a confidence interval for this trend was estimated as the interval containing $\alpha$ (i.e. 95%) of the sample points, for which the median is represented as the trend in Eq. 7.

To test the significance of the trend estimation, the Mann-Kendall test was used (Mann, 1945; Kendall, 1948). This method requires no specific distribution of the data, but must be applied to serially independent data (Collaud Coen et al., 2020). To account for this, prewhitening algorithms have been developed to reduce the influence of autocorrelation on the significance level of the derived trend. Such a prewhitening method is described in Yue et al. (2002), where the data is processed before performing the Mann-Kendall test. Following Yue et al. (2002), the estimated trend $T_{TS}$ was calculated using Eq. 7, before being removed from the time series $X_t$ and creating the detrended time series $X_t'$:

$$X_t' = X_t - T_{TS} \cdot t. \tag{8}$$

The lag-1 autocorrelation ($r_1$) was then calculated and used to produce the independent series $Y_t'$:

$$Y_t' = X_t' - r_1 X_{t-1}'. \tag{9}$$

The trend was then added to the independent series:

$$Y_t = Y_t' + T_{TS} \cdot t, \tag{10}$$

which was used to asses the trend significance using the Mann-Kendall test. The trend was deemed significantly different from 0 when the resulting p-value was less than $\alpha = 0.05$. Additionally, to study seasonality, the seasonal Mann-Kendall test (Hirsch et al., 1982) was used in order to acquire trends for autumn, winter and spring.



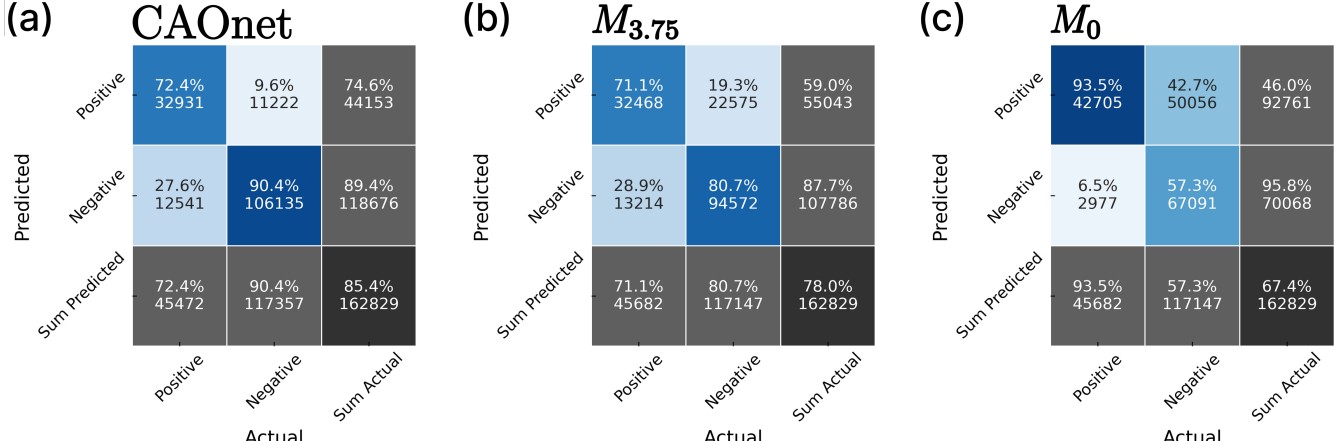

**Figure 3.** Confusion matrices for CAOnet (a), MERRA-2 with $M > 3.75$ K (b), and MERRA-2 with $M > 0$ K (c). The y-axis represents predicted classes by the models, while the x-axis represents the labeled classes. Positive corresponds to classified CAO, while negative means no CAO was classified. For each colored box, the lower number corresponds to number of classified patches, while the upper percentage corresponds to the rate of that predicted class relative to the actual class. The lower gray row shows the recall and true negative rate (upper) and number of patches labeled as that actual class (lower). The rightmost gray column shows the precision for positive and negative prediction (upper) and number of patches corresponding to that predicted class (lower). Finally the lower right corner in dark gray shows the total accuracy (upper) and total number of patches classified (lower).

## 3 Results and discussions

### 3.1 Model evaluation

The final configuration and structure of CAOnet was determined based on the model's accuracy. It was found that CAOnet performed best when it was implemented with an autoencoder-K-means combination with 7 clusters, of which 2 are defined as CAOs. This combination achieves an accuracy of 85.4%, a recall of 72.4%, a true negative rate of 90.4% and precision of 74.6%, as can be seen in Fig. 3a. Meanwhile when evaluating previously used CAO criteria for the $M$ index, it was found that

a threshold of $M > 3.75$ K ($M_{3.75}$) performed the best, in contrast to the much used threshold of $M > 0$ K ($M_0$). With $M_{3.75}$, an accuracy of 78%, a recall of 71.1%, true negative rate of 80.7% and precision of 59.0% is reached (see Fig. 3b), which is better than when using a threshold of $M_0$ reaching an accuracy of 67.4%, a recall of 93.5%, true negative rate of 57.3% and precision of 46.0% (see Fig. 3c).

The precision metrics highlight the strengths of CAOnet over the $M$ index. For $M_{3.75}$ the precision is only 59.0%, indicating

that close to half of all CAO predictions made are false positives. Furthermore, the reduction in precision to 46.0% for $M_0$, suggests that the majority of CAO predictions made using this threshold are false positives. This has significant implications for studies of CAO clouds employing the $M$ index. As $M > 0$ is frequently used, these findings emphasize the importance of selecting a higher threshold and considering how any chosen threshold influences the uncertainties of such analyses.



For further evaluation, visual inspection was performed on two typical CAO cases found when screening through the hand labeled dataset. Figure 4 shows one of these cases where CAOnet and the MERRA-2 $M$ index struggle to accurately predict CAOs, where disagreements with the evaluation dataset mostly show up as false negatives. $M_{3.75}$ in Fig. 4c appears to struggle with capturing the more open cellular structures (shown in orange), especially downwind in the lower right corner of the swath. By lowering the $M$ index threshold to $0\,\mathrm{K}$, these open cells are captured (see Fig. 4d), but with a slight increase in false positives (shown in red), suggesting that the threshold of $3.75\,\mathrm{K}$ may be too high to capture the open cells in this example.

Additionally, Fig. 4 shows an example where the labeler may have been too conservative in their labeling, underlining the issue of subjective bias. Region 1 in panels a and d reveals closed cell-looking clouds labeled by the $M$ index, but not by the human labeler. Similarly, region 2 in panel b and c shows what appears to be initial closed cell development, which has been labeled by the $M$ index but not by CAOnet or the human. This can be explained by the uncertainties associated with the subjectivity of human labeling. Stevens et al. (2020) showed that six individuals rarely reached unanimous agreement when labeling mesoscale shallow clouds in the trade winds. This illustrates how subjectivity can lead to an evaluation dataset that may not accurately reflect ground truth. Consequently, the classifications from a model such as CAOnet and MERRA-2 using the $M$ index is likely to provide more stable and accurate classifications than that produced by a single labeler. However, the evaluation dataset may still indicate which model is better at classifying easily distinguishable clouds, such as clear cases of closed and open cells that are easily distinguishable by a human. As a result, the higher accuracy of CAOnet over $M_{3.75}$ combined with a significantly higher true negative rate (90.4% vs 80.7%) and precision (74.6% vs 59.0%), suggests that CAOnet has captured more typical CAO clouds without the cost of more false positives that could bias its predictions.

Figure 5 shows a second classification example. Here, CAOnet (panel b) agrees well with the hand labeled data, while struggling to capture some of the initial closed cell development (marked as region 3). This is a general tendency of CAOnet seen in multiple classification examples throughout all seasons and years. It is a result of overlap between non-CAO cloud types and the cluster aligning with initial dense closed cell development. Discarding that cluster results in better overall accuracy, but at the cost of missing CAO classifications close to the sea ice edge. This limitation must be considered in all further analysis, as it may significantly impact the results. Especially, a shift in the prevalence of initial closed cell development off the sea ice edge as a result of climate change could result in lower or higher CAO detection over time, affecting the upcoming trend analysis. However, since the extent of initial cloud development is small compared to the total extent of a typical CAO, the correct classification of these clouds may be insignificant regarding the overall radiative impact of CAOs.

In Fig. 5c, $M_{3.75}$ also fails to detect some initial closed cell cloud development off the sea ice edge. While decreasing the $M$ index threshold down to $0\,\mathrm{K}$ solves some of the missing detection (see Fig. 5d), it also greatly increases false positives. One reason for these missed detections may be related to a typical shallow MBL of only a few hundred meters close to the sea ice edge (Fletcher et al., 2016b). As a result, the initial cloud formation may be a result of instabilities not stretching all the way to $850\,\mathrm{hPa}$ in MERRA-2. Consequently, a higher pressure level (lower altitude) would have to be used for more efficient cloud detection. However, as the large turbulent fluxes act on the MBL, the potential temperature at a lower altitude may no longer describe the presence of instability and CAO clouds. This suggests that relying on the potential temperature from a single pressure level like 700, 800 or 850 as used in previous studies (e.g. Kolstad and Bracegirdle, 2008; Fletcher et al., 2016b;




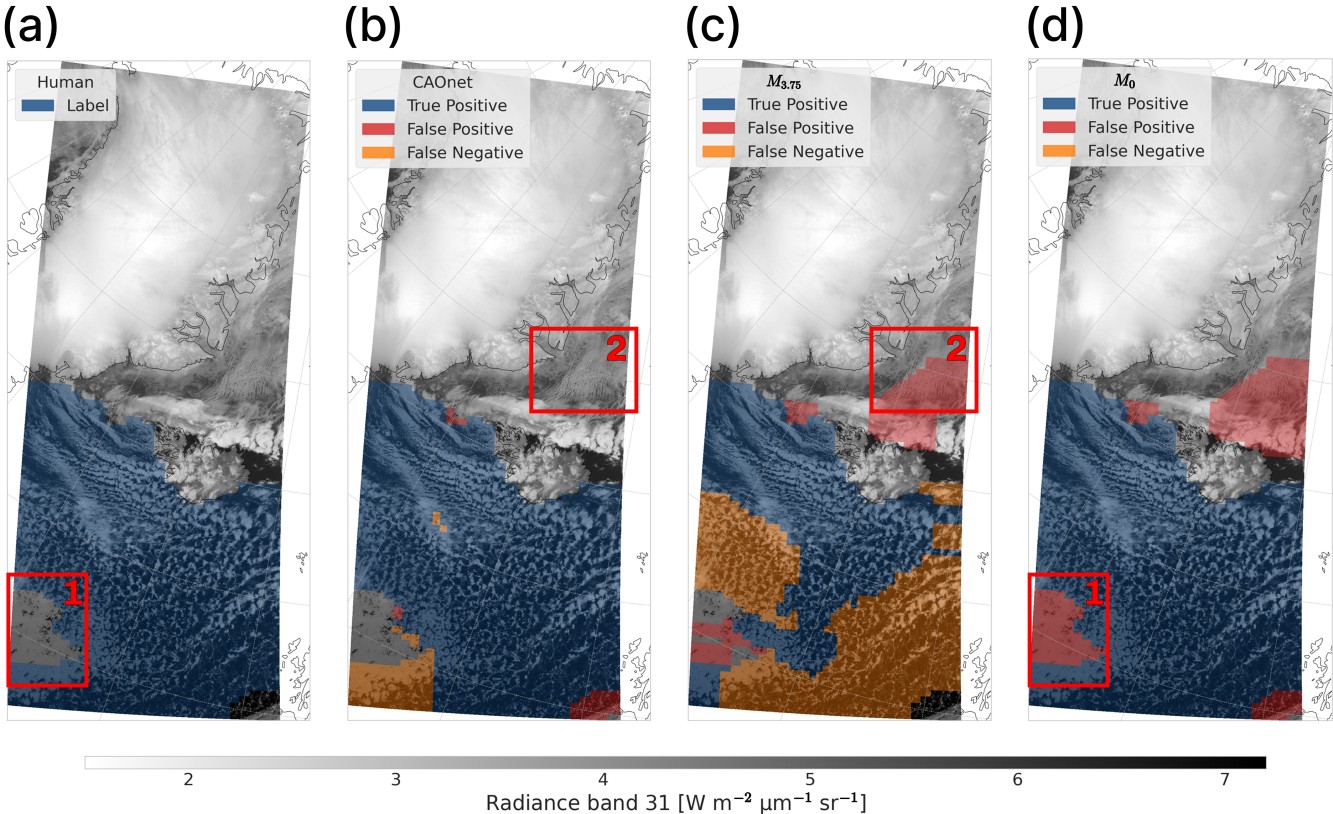

**Figure 4.** A CAO south of Iceland on 3 December 2013 22:50 UTC labeled by a Human (a), CAOnet (b) and MERRA-2 using a $M$ index threshold of 3.75 K (c) and 0 K (d). Regions 1 and 2 highlight possible CAO cases not detected by CAOnet or the human labeler.

Papritz and Spengler, 2017), may not be optimal for CAO cloud classifications using MERRA-2 or other reanalysis products
over the entire region where CAOs are found in the North Atlantic.

Additionally, the commonly used pressure levels for calculating the $M$ index account only for the lower troposphere, missing out on a potential upper troposphere cloud layer. This limitation can lead to $M$ index CAO classifications even when only high clouds are visible from space. Figure 5c, region 4 shows such an example where a high cloud extending from southern Norway along the CAO towards the sea ice edge to the northwest has been misclassified as a CAO. Although there may be CAO clouds
present below, their radiative influence is greatly reduced by the high cloud above. Moreover, the high cloud region is quite large, giving a considerable contribution to the CAO database produced by the $M$ index for this specific date. Even though an $M$ index threshold most closely aligning with the label data has been selected, such discrepancies often occur, indicating that careful consideration is required when aiming to use the $M$ index to study the radiative impact of CAO clouds.





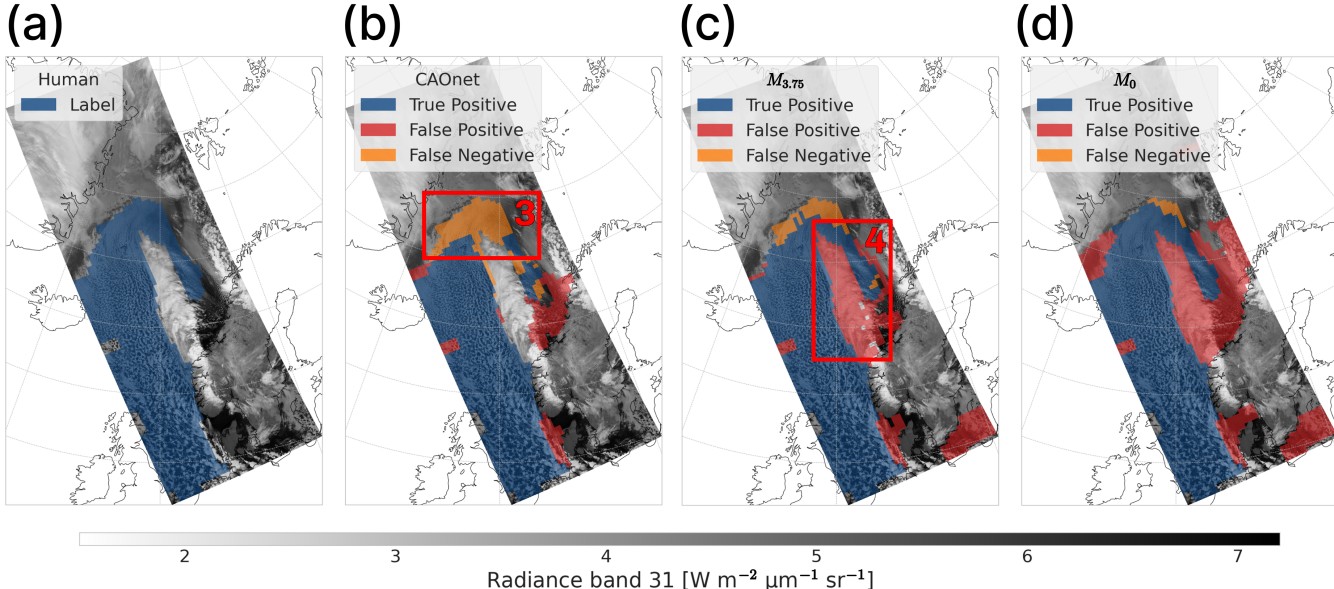

**Figure 5.** A CAO west of Norway 21 November 2008 21:00 UTC labeled by a Human (a), CAOnet (b) and MERRA-2 using a $M$ index threshold of $3.75\,\mathrm{K}$ (c) and $0\,\mathrm{K}$ (d). Region 3 shows a typical example of CAOnet struggling in capturing initial closed cell development, while region 4 shows an example of the $M$ index classifying a high cloud as a CAO.

## 3.2 Climatology

Having established the classification performance of CAOnet, $M_{3.75}$ and $M_0$ on individual images, they were applied to produce climatologies of CAO clouds for the months November to April from 2000 to 2025. In Fig. 6a, CAOnet shows RFO maxima close to 30% west of Lofoten along the Norwegian coast, and south of Iceland. These regions are located far from the sea ice edge and closely align with warmer SSTs (see Fig. A1b), where weaker CAOs are expected to be found (Papritz and Spengler, 2017). Similarly, a low RFO is observed in the Iceland and Greenland seas, where SSTs are expected to be relatively

low. While this suggests a CAO cloud SST dependency, these findings may be a consequence of the higher CAOnet detection rate of open cells and their formation farther from the sea ice where higher SSTs are found. The findings do however show an interesting SST-aligning pattern, which suggests higher SSTs as a potential necessity for CAOnet detection and thereby open cell formation.

The Fram Strait is a region known for CAOs due to frequent instability in the MBL (Papritz and Spengler, 2017; Dahlke

et al., 2022). However, this region shows relatively low RFO for CAOnet in Fig. 6a. This discrepancy may be attributed to earlier studies employing the $M$ index which suggests instability without necessarily indicating cloud formation. It could also result from high clouds obscuring the CAO clouds below or from CAOnet failing to capture the initial development of closed cell clouds, as shown in Fig. 5. In contrast, $M_{3.75}$ and $M_0$ aligns with expectations, showing maxima in the RFO over the Fram





Strait (Fig. 6 panel b and c). This maximum also extends towards the Norwegian coast, giving higher RFO in this region than
335 CAOnet.

The Norwegian coast is a region where closed cells are expected to start breaking up into open cells. As illustrated in Fig. 4
and 5, CAOnet shows high sensitivity to open cell clouds, suggesting that CAOnet likely represents their RFO of approximately
30%. Consequently, $M_{3.75}$ overestimates this occurrence with an RFO of more than 35% . Additionally, the low $M_{3.75}$ CAO
precision of just 59% (see Fig. 3b), implies a large number of false CAO classifications. These inaccuracies for $M_{3.75}$ may
mainly result from high clouds obscuring the CAO clouds below (i.e. region 2 in Fig. 5c), giving non-aligning MERRA-2
classifications. While this is not necessarily wrong, such cases are less relevant when studying the radiative impact of CAO
clouds.

Another region of disagreement between CAOnet and the $M$ index is south of Iceland, where CAOnet (Fig. 6a) shows up
to 20% higher RFO than $M_{3.75}$ (Fig. 6b). Given the CAOnet precision of 74.6% (Fig. 3a), it is unlikely that this is the result
of large amounts of false positives. Instead, it may be explained by lacking CAO classifications by the $M$ index, which is
shown as an example in Fig. 4. This Figure illustrates that the region south of Iceland can be dominated by open cells, which
MERRA-2 may fail to detect, unless the $M$ index threshold is lowered (see Fig. 6c).

By decreasing the $M$ index threshold to 0, the CAO occurrence south of Iceland increases by approximately 30%, but
this comes at a significant cost to overall precision. With a precision of only 46.0% (see Fig. 3c), most of the $M$ index CAO
predictions using $M_0$ are false positives. This shows how lowering the $M$ index threshold can be beneficial in certain cases such
as in Fig. 4, but with the consequences of lowering the total accuracy. This highlights the strength of the phenomenological
approach using CAOnet as a CAO cloud predictor, provided that its limitations regarding missing classifications of initial
closed cell clouds are acknowledged.

To improve classifications by MERRA-2 in regions where open cells are expected, it may be beneficial to use another
355 pressure level when calculating the $M$ index. Near the sea ice edge, CAOs and the MBL depth are expected to be a few
hundred meters, increasing to up to 2 km downstream (Fletcher et al., 2016b). As a result, higher pressure levels may lead to
missing classification close to the sea ice edge, making the 850 hPa potential temperature more suitable for detecting general
instability and the potential for convective clouds. However, as open cells develop, the MBL may further deepen through
mixing, suggesting that a lower pressure level might be better suited for detection of these open cells. Implementing a varying
pressure level or $M$ index threshold based on region and CAO lifecycle could address these issues, but would require extensive
research to determine accurate parameters.

In total, the climatology from CAOnet (Fig. 6a) identifies regions where CAO clouds play a significant role for the radiative
balance. This offers an alternative perspective to previous studies, focusing on MBL instability associated with CAOs. While
instability remains an important factor for exchange of heat fluxes (Papritz and Spengler, 2017), CAOnet demonstrates its
strengths in applications directly aimed at studying CAO clouds. Other than potentially missing important areas like the Fram
Strait and close to the sea ice edge, the identified areas of high RFO point to the coastal Norwegian Sea and the northern
Atlantic south of Iceland as key regions for investigating the radiative impact of CAO clouds.



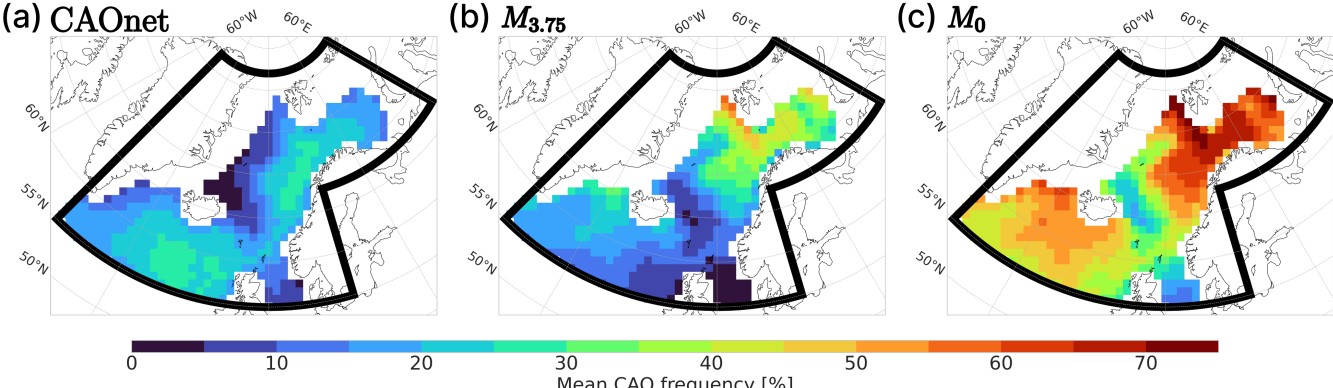

**Figure 6.** CAO climatologies for CAOnet (a) and the $M$ index calculated using MERRA-2 reanalysis for the months November-April using data from March 2000 to February 2025. In panel b and c, $M$ index thresholds of 3.75 and 0 K were used respectively. Regions with less than 90 % data availability because of missing MODIS swaths or sea ice have been discarded.

Supplementing the RFO climatology, the average CAO cloud coverage for each month is investigated. As visualized in Figure 7a, the average CAO coverage peaks during the winter months for each model. This aligns with RFO maxima reported

in earlier Arctic CAO studies (e.g. Kolstad et al., 2009; Dahlke et al., 2022), suggesting that both CAOnet and the $M$ index effectively capture the expected CAO seasonality. When further comparing the monthly coverage, the results from CAOnet and $M_{3.75}$ mostly agree. This is expected, as both the $M$ index threshold and CAOnet have been optimized for the best fit with the evaluation data. Nevertheless, discrepancies are still observed. While both models show similar coverage in winter and spring, they disagree in autumn, with $M_{3.75}$ showing lower CAO coverage. This discrepancy may indicate that $M_{3.75}$ has a

too high threshold to detect clouds associated with weaker CAOs, which may occur when Arctic air masses are warmer during autumn. $M_0$ yields more comparable results to CAOnet for September, but the adjustment leads to a doubling of coverage in the following months. The alternative offered by CAOnet suggests that varying the $M$ index thresholds by season may be necessary to optimize an $M$-index-based CAO cloud detection.

Despite the disagreements between the models, the clear seasonal pattern obtained suggests a corresponding pattern in terms

of the cloud radiative impact. With maximum coverage occurring during winter and early spring, it is expected that these months will experience the most significant CAO cloud radiative impacts. Consequently, December to March are likely to play an important role in CAO influence on the Arctic climate, especially if trends in coverage are present. To further explore this, these months will be emphasized in the following sections, with a focus on CAOnet, while utilizing supporting results from MERRA-2 with $M_{3.75}$.





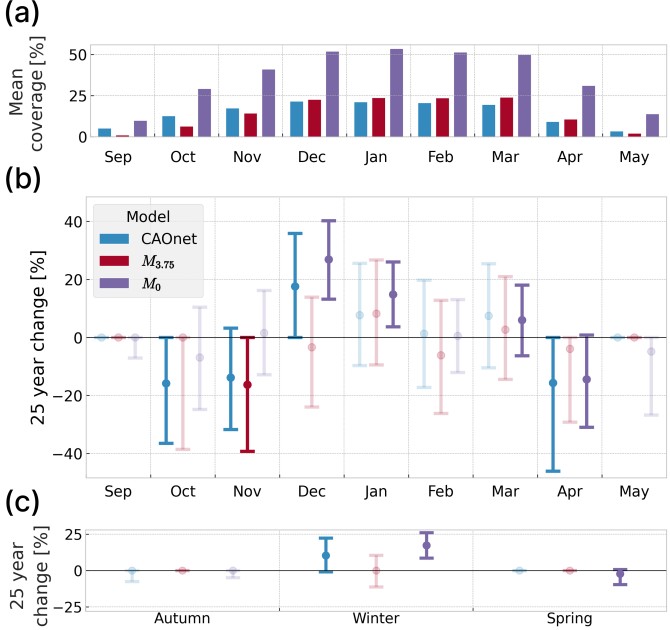

**Figure 7.** Trends in CAO coverage for September to May, 2000 to 2025 calculated using CAOnet (blue) and MERRA-2 with a $M$ index threshold of 3.75 K (red) and 0 K (purple). Panel (a) shows the mean coverage of CAOs for the whole 25-year period. Panel (b) shows the relative 25-year change in CAO coverage for each month while panel (c) shows the relative 25-year change in CAO coverage for each season. Error bars indicate the 95 % confidence interval of the Theil-Sen slope, with the Theil-Sen estimate shown as dots. Significant trends are shown as solid colors, indicating a p-value less than 0.05 estimated using the non-parametric Mann-Kendall test. Insignificant trends are shown as transparent colors.

## 3.3 Trends

### 3.3.1 Arctic CAO cloud cover trends

To investigate how CAO clouds are responding to a warming Arctic, trends in their coverage are analyzed. Figure 7b shows trends in CAO cloud coverage for September-May as a 25-year relative change for CAOnet and the $M$ index calculated using MERRA-2. CAOnet suggests a seasonal trend in which the shoulder months of the CAO season (October, November and April) show a significant relative decrease in cloud coverage of nearly 20 %, in contrast to the winter season and March that show increases of approximately 10 % and almost 20 % respectively. While the winter increase is not significant on a month-by-month basis, except for March, the nearly 10 % total winter season trend in Fig. 7c is significant. As the winter also contains the highest CAO coverage (see Fig. 7a), this relative trend has a great impact on the total area covered by CAO clouds.

To further support these findings, the trends obtained using the $M$ index can be examined. In Fig. 7b, $M_{3.75}$ generally agrees with the sign of the CAOnet trends, except for December, when CAOnet suggests an increase and the $M$ index a significant decrease. However, the confidence interval of this trend is well inside the confidence interval of CAOnet, suggesting a non-



significant disagreement. When the $M$ index threshold is reduced to $0\,\mathrm{K}$, the trend sign shifts to a strong and significant increase. However, as explored in Sect. 3.1, this threshold does not typically align well with CAO clouds. Nevertheless, the significance of the trend suggests an increase in areas with MBL instability for December across the Arctic. Overall, the
400 seasonality of the MBL instability and cloud cover trends suggests clear seasonal drivers, which may impact both the exchange of heat fluxes and the radiative balance of the Arctic.

### 3.3.2 Explaining the seasonality of Arctic trends

The observed seasonality of the trends may be explained by Arctic amplification and trends in surface air and sea surface temperatures. As CAOs and their associated clouds are dependent on MBL instability, Arctic amplification may reduce their
coverage through rising air temperatures. While SSTs are also increasing (Garcia-Soto et al., 2021), this process is much slower as visualized in Fig. A1, leading to a dominating air temperature effect on the MBL instability. This may especially influence the shoulder seasons when the contrast between sea and air temperature is lower, providing insufficient instability for the initial development of clouds, explaining the observed trends in Fig. 7b. For instance, if a minimum $M$ index of $3\,\mathrm{K}$ is generally required for CAO cloud formation, seasons that were already close to this threshold at the beginning of the century are likely to
see decreasing cloud formation given Arctic amplification, in contrast to seasons that typically experience greater instabilities.

Additionally, vertical profiles of Arctic warming may help explain the observed trend seasonality. Arctic air temperature warming is observed to be greater (Chylek et al., 2009; Johannessen et al., 2016; Rantanen et al., 2022) and more surface-confined during winter than in the shoulder seasons (Graversen et al., 2008; Alexeev et al., 2012). This makes the amplified wintertime warming a destabilizing factor for the MBL, in contrast to the more vertically distributed warming during the
415 shoulder seasons. This helps explain the increasing prevalence of CAO clouds during winter and their decreasing prevalence during the shoulder seasons, as observed in Fig. 7.

Other processes such as changes in SSTs, sea ice retreat or circulation patterns may also contribute to the observed trends. In Fig. 6, CAOnet suggests higher RFO of CAO clouds in regions with higher SSTs, revealing an SST dependence. In the Tropics, Sandu and Stevens (2011) showed that closed-to-open cell transition occurred as a result of increasing SSTs. Consequently, as
SSTs rise due to climate change, conditions may become more favorable for cumulus and open cells, which are easily detected by CAOnet. Following this, increasing wintertime CAO cloud cover could be attributed to increasing prevalence of open cells.

Large-scale weather patterns may also influence the coverage trends of CAOs. Kolstad et al. (2009) found a significant correlation between the $M$ index and the North Atlantic Oscillation (NAO). While the NAO influences the Arctic regions differently, the southern region has a positive correlation with CAOs, suggesting that trends in CAO coverage may be caused
by changes in the NAO. However, (Landgren et al., 2019) found that projected changes in CAO occurrence were dominated by changes in temperatures and sea ice and not circulation patterns. Together with temperature changes, Arctic sea ice retreat may therefore be the main causes of the observed seasonality and trends.

The loss of Arctic sea ice in recent years, which is projected to continue into the future (DeRepentigny et al., 2016), is unlikely to explain the seasonality of the observed trends. Specifically, observation indicate a larger wintertime sea ice loss
(Garcia-Soto et al., 2021), which leads to a northward shift of the sea ice edge, thereby affecting cloud development and poten-



tially break-up. This northward shift in the onset of break-up is expected to result in northward wintertime cloud dissipation, contributing to a reduction in CAO cloud cover in the study domain, which is not observed in the trends. While the exact impact of sea ice remains uncertain, it underlines the need for further investigation to better understand future development of CAO clouds as sea ice continues to retreat.

### 3.3.3 Radiative impacts of Arctic trends

Low level clouds contribute the most to the Arctic surface radiative balance (Shupe and Intrieri, 2004), making trends in the low level CAO clouds of high importance for Arctic warming. Their net radiative effect, do however largely depend on solar radiation, which is negligible during winter for large portions of the study region. This results in a dominant longwave radiative warming from the increasing coverage of the CAO clouds during winter, which may have contributed to an enhanced and region-dependent wintertime Arctic amplification (Rantanen et al., 2022). In contrast, during the shoulder months, solar radiation contributes significantly to the radiative balance, which may lead to a warming effect associated with the decreasing prevalence of CAO clouds. This could be offset by the increasing longwave surface radiative cooling from the decreased CAO cloud coverage, although the exact radiative impact remains uncertain.

Additionally, changes in CAO cloud cover potentially align with shifts in other cloud types, complicating their radiative impact. This underscores the importance of studying the atmospheric conditions responsible for the observed CAO trends. However, as the $M$ index based on MBL properties supports the trends found using CAOnet, it is plausible that the trends arise from local MBL changes or circulation patterns influencing the MBL. Consequently, trends in clear sky or other MBL clouds are likely correlated with changes in CAO clouds, with CAO-replacing MBL clouds minimizing, and clear sky maximizing the radiative impact of these trends.

While we can expect the observed changes to have had a net warming effect, large uncertainties remain and further investigation is needed. Future studies could explore the relationship between changes in CAO cloud cover and the atmospheric phenomena that are potentially replacing them. Additionally, satellite radiative budget products could be applied to regions where changes are observed. This could uncover an exact radiative impact of the observed CAO trends, and provide insights into how Arctic maritime clouds may evolve in the future.

### 3.3.4 Trends and coverage in the north and south

To better understand changes in the radiative impact of CAO clouds, it is important to study both seasonal and latitudinal trends. As the North Atlantic ocean stretching to the Arctic maintains fairly stable temperatures throughout the year (Fig. A1), the outgoing longwave radiation from the ocean surface remains close to constant. However, the incoming solar radiation may vary greatly, particularly in the winter season across the domain between 55 and 80 °N. This latitudinal dependency motivates a separate trend analysis for the southern and northern part of the domain (see Fig. 2).

Figure 8 shows the trends and CAO cloud coverage for the northern (panels a-c) and southern regions (d-f). Similar to the whole domain in Fig. 7, the highest CAO coverage occurs in both regions during winter. However, there is a shift in maximum coverage towards late winter and spring in the northern region (a) for both the $M$ index and CAOnet, while in the



southern region CAOnet shows a shift towards early winter and autumn (d). In the North, the $M$ index generally suggests more
CAO coverage than CAOnet, whereas CAOnet suggests higher coverage than the $M$ index in the South. This aligns with the
climatology presented in Fig. 6, where the $M$ index shows higher occurrences than CAOnet in the north and lower in the south.

In general, a higher occurrence of CAOs is expected farther north as a result of CAO initiation near the sea-ice and snow-
covered surfaces, which are more prevalent in the north. However, as these clouds move over the ocean toward the south,
they develop into open cells that may expand to cover large areas. Additionally, CAOs originating outside of the study area,
specifically west of Greenland, can extend into the southern region as visualized in Fig. 5. This results in a higher cloud
coverage in the South than the North, despite the expectation of higher CAO occurrence in the North. Consequently, the high
cloud coverage indicated by CAOnet in the South (Fig. 8d), along with CAOnet demonstrating high sensitivity to open cell
clouds in this region (Fig. 5b), enhances confidence in the results produced by CAOnet. This motivates the importance of the
southerly trends (Fig. 8 panels e and f), especially when interpreting these findings as mostly reflecting changes in open cell
coverage.

While $M_{3.75}$ sees a significant decrease in February, no general trend in CAO coverage is observed for CAOnet in the
northern region (panels b and c). In contrast, $M_0$ reveals significant increasing trends for December and January and decreasing
for February, as well as an increasing trend for the whole winter season combined. This suggests an increase in areas with MBL
instability over the last 25 years, with corresponding insignificant trends in cloud cover as indicated by CAOnet. This can be
explained by the $M$ index trend contributions largely resulting from increasing areas of very weak instability, insufficient to
form clouds.

In the southern region (Fig. 8 panels e and f), a seasonal pattern is observed, marked by an increase in cloud cover during
winter and decrease in the shoulder seasons autumn and winter. This pattern aligns with the seasonality of the whole domain
(compare Fig. 7b), suggesting that the trends in the southern region are driving those observed over the entire domain. This
indicates that changes in the prevalence of open cell clouds, which are frequently found in the South, are the main contributors
to the observed trends. Notably, significant increases in cloud cover are found in December (Fig. 8e) and the winter season
combined (Fig. 8f), while a significant decrease is found in the shoulder months October and November (Fig. 8e). $M_0$ exhibits
similar significant trends for October and December, while also suggesting a significant cloud cover decrease decrease for
April. In contrast, $M_{3.75}$ suggests no overall trend, deviating from the two others and indicating that the $M$ index threshold
may be too high to capture the cloud structures typically found in the southern region.

The overall trends attributed to the southern region and its open cell coverage suggest potential changes in atmospheric
circulation and the efficiency of cloud dissipation over the past 25 years. While shifts in circulation patterns may result in
more or fewer CAO clouds reaching the South, these trends could also be caused by delayed or enhanced dissipation of
open cell clouds. Regardless of the underlying cause, the increasing prevalence of open cells during winter and decreasing
prevalence during the shoulder seasons will change the radiative impact of CAOs. Although these factors only provide plausible
explanations for the observed trends, they emphasize the importance of understanding both circulation changes and cloud
dissipation within CAOs, as these processes may themselves be influenced by climate change.



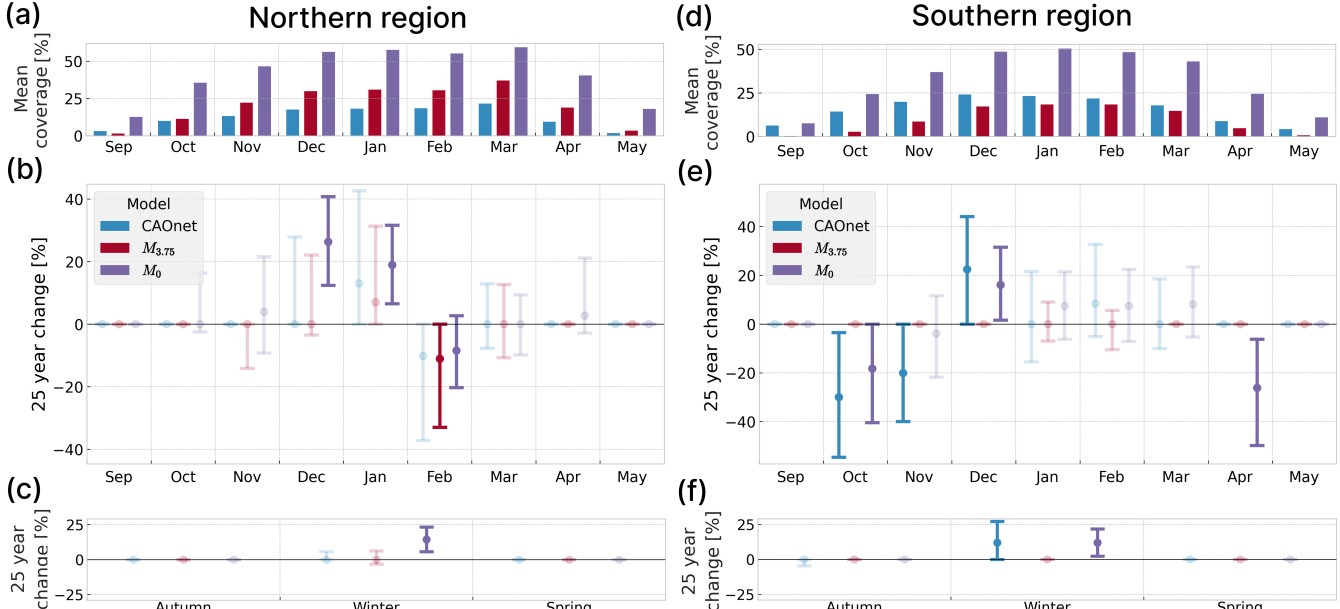

**Figure 8.** Similar to Fig. 7, but for the northern region north of 65 °N (a-c), and the southern region between 55 and 65 °N (d-f) as shown in Fig. 2. Significant trends are shown as solid colors, indicating a p-value less than 0.05 estimated using the non-parametric Mann-Kendall test. Insignificant trends are shown as transparent colors.

Another explanation for the wintertime trends is the increased availability of energy and moisture from the higher SSTs in the North Atlantic, which may feed and sustain the convective open cells, counteracting water loss through precipitation. This influences the radiative impact of the clouds in terms of both increased cloud lifetime and higher optical depth. Although the resulting increase in cloud albedo has a minimal impact during the winter months, it may help counteract the terrestrial warming effect of the increased cloud coverage by increasing the reflection of solar radiation to Space in Autumn and Spring. The exact balance between the solar and marine longwave effects remains uncertain and warrants further investigation, especially as trends are also observed in the shoulder seasons, when the solar radiative effects may become significant. Consequently, further studies could utilize a phenomenological CAO classifier like CAOnet to directly assess changes in radiative effects of regions experiencing these trends. This approach could provide more accurate insights into the future impacts of Arctic CAOs on both the local and global climate.

## 4 Conclusion

Clouds associated with CAOs are important for the Arctic radiative budget, particularly in the regions surrounding the Norwegian sea, Barents sea and Northern Atlantic due to their extensive coverage. In the rapidly warming Arctic, it becomes important to understand how these clouds respond to climate change, as changes in their prevalence and properties may ei-





ther amplify or dampen local and global warming. To explore shifts in CAO cloud coverage, a phenomenological CAO cloud classification method named CAOnet has been developed. This approach based on MODIS data and machine learning has demonstrated promising results compared to the traditional marine cold air outbreak index ($M$) in detecting closed and open cell clouds. Additionally, an $M$ index threshold of 3.75 K has been identified for optimal detection of CAO clouds, providing future studies a basis for the instability required for CAO cloud formation.

By employing CAOnet, a CAO climatology focusing on clouds rather than MBL instability has been produced. In contrast to the $M$ index calculated using MERRA-2 reanalysis, this climatology has provided an alternative perspective on CAOs, highlighting the Norwegian coast and the North Atlantic region south of Iceland as key areas for CAO clouds and their associated radiative impact. The correlation of these regions with relatively high SSTs, as well as frequent open cells to which CAOnet shows a high sensitivity, suggests that SSTs may play a role in the formation and maintenance of open cell CAO clouds.

Utilizing data from the past 25 years, the analysis of cloud cover trends has revealed a 10 % increase in wintertime CAO cloud cover and nearly a 20 % decrease during the shoulder months of October, November and April. These shifts are in large part linked to changes in open cell cloud cover as well as changes in the MBL (as indicated by the $M$ index). Specifically, the trends are driven by faster warming of the air compared to the sea, which leads to reduced instability.

During the shoulder months, this reduced instability is enough to disturb initial cloud formation, while changes in already strong wintertime instabilities do not affect cloud formation. A more surface confined atmospheric warming profile during winter is thought to further amplify the seasonality of the trends through enhanced wintertime instability and cloud formation. Additionally, rising SSTs may result in conditions favorable for open cell formation during winter, leading to an increased detection of CAOs by CAOnet. This is supported by the wintertime trends being mostly evident in the southern Arctic domain (55 to 65 °N), where SSTs are higher and open cell clouds more prevalent. This suggests increasing SSTs and thereby potentially more open cells as a driver of the observed wintertime trends.

Due to the lack of incoming solar radiation during winter, the radiative impact of the observed wintertime cloud cover trends are dominated by the terrestrial radiative warming effect. This may have contributed to the anomalously strong and region dependent wintertime Arctic amplification. Conversely, during the shoulder months when decreasing cloud cover is observed, the increased solar radiative effect introduces uncertainties regarding the overall radiative effect of the cloud cover trends.

While these trends do not directly indicate future changes in the Arctic radiative balance, they clearly indicate that CAO clouds are influenced by climate change. This emphasizes the importance of further research in order to accurately characterize the radiative impact of these clouds and understand their role in the local and global climate system.

*Code and data availability.* Data, including CAOnet and MERRA-2 CAO masks, along with the code, are available from Zenodo at https://doi.org/10.5281/zenodo.16680335 (von der Lippe, 2025). MODIS level 1B calibrated radiances can be accessed at https://ladsweb.modaps.eosdis.nasa.gov/search/order/1/MOD021KM--61 (MODIS Characterization Support Team (MCST), 2017). NIMBUS sea ice concentration data is available at https://nsidc.org/data/nsidc-0051/versions/2 (DiGirolamo et al., 2022). The MERRA-2 reanalysis products are accessible at https://disc.gsfc.nasa.gov/datasets/M2T1NXSLV_5.12.4/summary (Global Modeling and Assimilation Office (GMAO), 2015)




# Appendix A: Sea surface temperatures

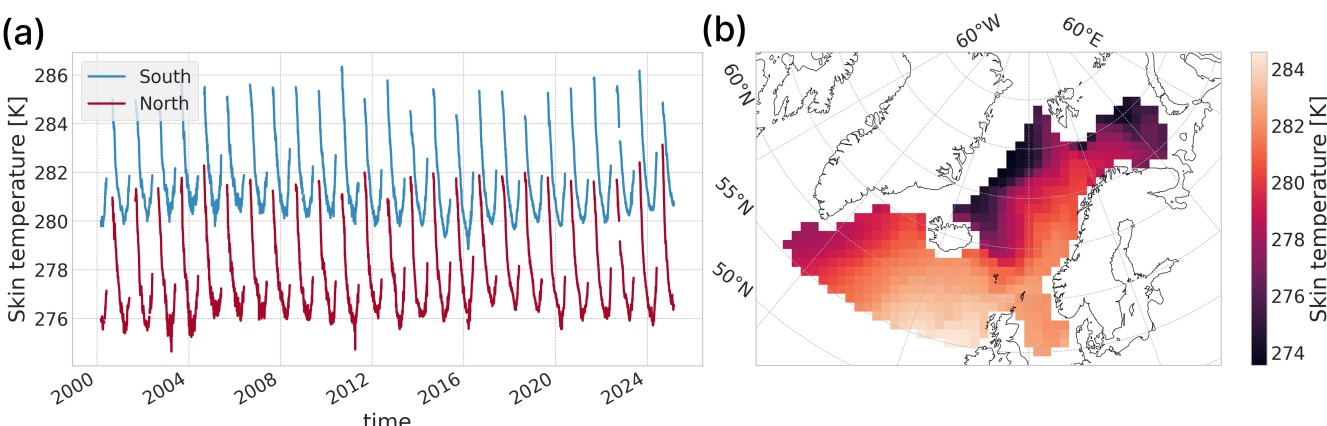

**Figure A1.** Surface skin temperatures in Kelvin. Panel a shows variations in the average temperature for the southern and northern region for the months September through May, while panel B shows the average skin temperature over the whole period.

# Appendix B: Optimizing $M$ index threshold

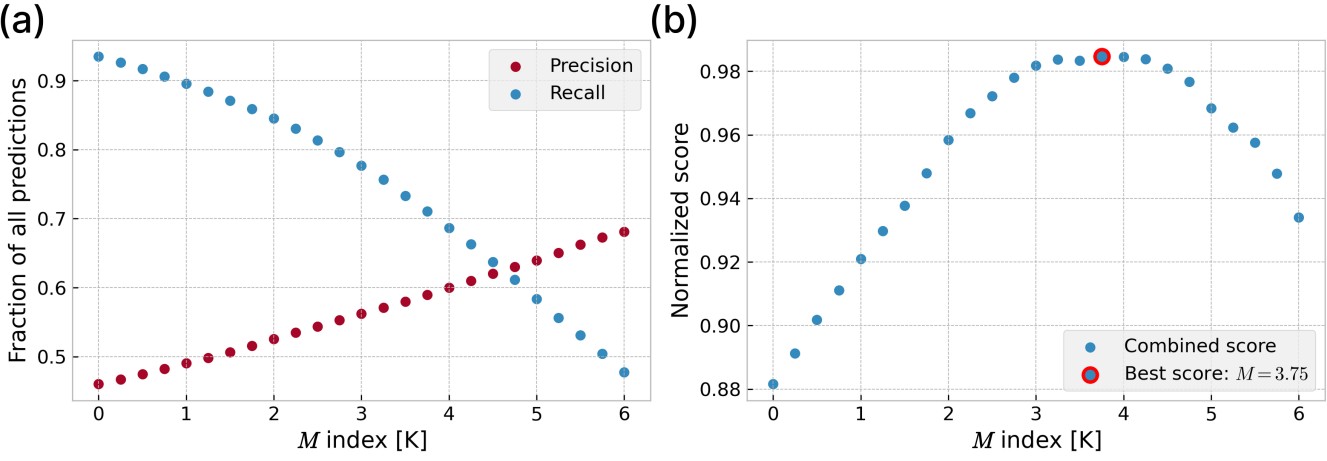

**Figure B1.** CAO evaluation scores calculated using different $M$ index thresholds. In panel a, a higher precision comes at the cost of recall. This leads to the the most optimal threshold of 3.75 K for the normalized score in panel b.



## Appendix C: Manual inspection of CAO images



**Figure C1.** Example images of CAOs used to identify CAO-aligning clusters to facilitate the creation of a labeling dataset. Out of the total 14 K-means clusters, manual inspection identified clusters 2, 6, 9 and 10 to align with typical CAO clouds.



*Author contributions.* FSVDL, ROD, and TC designed and conceptualized the study. FSVDL conducted the labeling, formal analysis, investigation and developed the methodology with supervision from ROD, TC and TS. The manuscript was written by FSVDL with contributions from ROD, TC and TS.

*Competing interests.* The authors have declared that none of the authors have any competing interests.

*Acknowledgements.* This research was supported by the European Research Council through ERC Consolidator Grant STEP-CHANGE Grant 101045273 and EU-HORIZON-WIDERA-2021 Grant 101079385 (BRACE-MY). We acknowledge the use of imagery from the NASA Worldview application (https://worldview.earthdata.nasa.gov), part of the NASA Earth Science Data and Information System (ES-DIS). We are also grateful to NRIS Sigma2 computing resources. We thank Franz von der Lippe for developing a satellite data labeling interface. Although generative AI was not directly involved in the writing of this manuscript, we acknowledge its use in suggesting language formulations.





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
