# Peer review of "Shortening of the Arctic cold air outbreak season detected by a phenomenological machine learning approach"

_EGUsphere, 2025_

## Author Response (AR1)

**Response to reviewer comments**

Filip Severin von der Lippe, Tim Carlsen, Trude Storevlmo, and Robert Oscar David

October 2025

We would like to thank the reviewers for their comments and suggestions, which have greatly improved the manuscript. Their comments are displayed below in bold, followed by our responses and the corresponding adapted manuscript text in italics. All figures, except for Fig. B1 and B2, are found in the adapted manuscript, where they retain the same figure numbering. All line numbers in our responses refer to the final manuscript.

**Reviewer 1**

**1)**

**The discussion of processes concerning open cells may be lacking. Particularly on lines 37-38 when open cells are tied to drying processes. This is somewhat contradicted by Eastman et al. (2022) that shows the closed-to-open Sc transition associated with increased boundary layer moisture and stronger fluxes (particularly surface winds and precipitation), in contrast to the transition to more disorganized cloud types, which are associated with drying. Increased SST is likely associated with stronger fluxes, so an SST-driven mechanism is still probable here, but the mechanics are unlikely due to drying processes.**

Thank you for the comment and reference. The idea of drying here was related to the dissipation of the stratocumulus layer, which can occur even if with increasing surface fluxes and more moisture. This stems from studies such as Abel et al., 2017, which suggest that precipitation and decoupling dries the stratocumulus cloud layer, switching to a regime dominated by cumulus. We do however see how the text in the paper may be misleading by talking about "drying of the cloud layer". Therefore, we have adjusted the text on lines 39-45 in the revised manuscript, to make this clearer and included the additional reference:

> *Several studies have suggested the onset of precipitation as a main driver for cloud break-up (e.g., Abel et al., 2017; Tornow et al., 2021; Yamaguchi et al., 2017). This precipitation, combined with increased winds, may further aid break-up into open cells by increasing the MBL moisture (Eastman et al., 2022), and favor the formation of cumuliform clouds (Stevens et al., 1998). Additionally, precipitation and evaporative cooling of the lower MBL (Abel et al., 2017) may lead to a decoupling of the stratocumulus cloud layer from the moisture-supplying surface (Bretherton & Wyant, 1997), contributing to the break-up. Furthermore, other research suggest that changes in MBL stability (McCoy et al., 2017), such as from increasing sea surface temperatures (SSTs), also play a role in this process.*

**2)**

**Throughout the manuscript, there is discussion of trends specifically for open cells (for example, line 485). This seems a bit speculative, since the CAO dataset does not appear to directly assess cellular structure, unless I am misinterpreting something. It may be wise to tone down these assumptions, since changes in MCC structure aren't really being assessed here.**

[Figure]

Figure 5: A CAO west of Norway 21 November 2008 21:00 UTC labeled by a Human (a), CAOnet (b) and MERRA-2 using a $M$ index threshold of $3.75\,\mathrm{K}$ (c) and $0\,\mathrm{K}$ (d). Region 3 shows a typical example of CAOnet struggling in capturing initial closed cell development, while region 4 shows an example of the $M$ index classifying a high cloud as a CAO.

Thank you for the comment. Although the dataset is created based on recognition of cellular cloud structure, it is true that it makes no clear distinction between closed and open cells. Nevertheless, open cells are still likely the largest contributor to the observed trend due to the following:

First, the CAO dataset has shown higher sensitivity to CAO-related open cells compared to closed cells, leading to some bias in the dataset towards open cells. The trends may therefore be more representative of CAO-related open cells than all CAO clouds, although this is not certain, as the model makes no such distinction. On lines 317-321 we describe why CAOnet misses closed cells, leading to it classifying more of the open cells.

> *Figure 5 shows a second classification example. Here, CAOnet (panel b) agrees well with the hand labeled data, while struggling to capture some of the initial closed cell development (marked as region 3). This is a general tendency of CAOnet seen in multiple classification examples throughout all seasons and years. It is a result of overlap between non-CAO cloud types and the cluster aligning with initial dense closed cell development. Discarding that cluster results in better overall accuracy, but at the cost of missing CAO classifications close to the sea ice edge.*

Second, the open cell trend attribution is not based on the direct observation of open cells, but the expectation of more open cells being present in the southern region, where clearer trends are found. To make this clearer we have updated the manuscript on lines 255-259:

> *The northern region was chosen based on typical CAO trajectories, capturing CAOs earlier on in their development due to the proximity to the sea ice edge. This typically results in larger concentrations of closed cells. In contrast, the southern region was chosen for its distance from the sea ice edge, capturing CAOs later on in their development (Murray-Watson et al., 2023), where a greater prevalence of open cells are expected (Brümmer, 1999).*

Third, to further underline the uncertainties related to the open cell trend attribution, we have updated the manuscript on lines 510-512 to motivate future work.

> *Additionally, by further developing this method to assess closed and open cells separately, closed*

*and open cell trend attribution may be determined accurately. This would yield more precise insights into the future impacts of Arctic CAOs on both the local and global climate.*

**3)**

**Line 390: I don't see a 20% increase in CAOnet in March, but I do see one in December.**
**Line 395: I also don't see any significant decreasing trends for any winter month for any of the CAO measures. It is possible that I do not have a current version of this figure in my manuscript, but the discussion of Figure 7 does not appear to line up with what I'm seeing on the figure.**

Thank you for catching this. The figure was updated last minute with one additional year of data, and some of the text had not been updated accordingly. The addition slightly altered the p-values for March and December, resulting in the December trend becoming significant and the March trend becoming insignificant at a 0.05 level. The text is now updated by replacing "March" with "December" on line 416 and line 418 to reflect the new figure.

**4)**

**As far as attribution of trends is concerned, much of this appears entirely speculative. In fact, the authors do not show any correlation analysis between time series of their CAO data and SST or ice edge data, which may strongly aid any discussion of attribution. I recommend revamping this section in order to more clearly show these proposed relationships using your new CAOnet data, or changing it a bit to motivate future work to do this, while not attempting to attribute the trends to anything here. Additionally, are any other flux variables known to be changing in this region? Trends in wind speeds or direction may be interesting.**

Thank you for the suggestion. We agree that the attribution section was rather speculative. To improve this, we have completely revamped this section to motivate future work to investigate trend attribution. Additionally, we have added an SST-CAO correlation figure (A1a), and an SST trend to Fig A2 to directly show how little SSTs are changing in this region. While no large correlation or trend is found, we motivate future studies to look into this as more data becomes available. The manuscript was updated on lines 431-453 to reflect this.

[Figure]

Figure A1: Panel a shows the SST-CAO correlation, with positive values indicating higher CAO occurrence for higher temperatures, while panel B shows the average skin temperature over the whole period.

[Figure]

Figure 6: CAO climatologies for CAOnet (a) and the $M$ index calculated using MERRA-2 reanalysis for the months November-April using data from March 2000 to February 2025. In panel b and c, $M$ index thresholds of 3.75 and 0 K were used respectively. Regions with less than 90 % data availability because of missing MODIS swaths or sea ice have been discarded.

[Figure]

Figure A2: Variations in the average temperature for the southern and northern region for the months September through May, including the significant Theil-Sen trend estimate.

*The observed trends motivate future studies to investigate the climatological parameters driving them. Especially, the seasonal nature of these trends underscores the potential importance of other season-dependent parameters such as temperatures and sea ice extent. Through CAOnet, future studies have access to a phenomenological classifier that provides an extensive database for correlation analysis with potential CAO-influencing climatological parameters. To aid future research in their analysis, we propose several factors for further investigation in order to better understand CAOs in a future Arctic.*

*First, Arctic air temperature warming tends to be greater (Chylek et al., 2009; Johannessen et al., 2016; Rantanen et al., 2022) and more surface-confined during winter than in the shoulder seasons (Alexeev et al., 2012; Graversen et al., 2008). This may serve as a wintertime destabilizing factor for the MBL, in contrast to the more vertically distributed warming during the shoulder seasons. However, increasing air temperatures close to the ocean surface may also limit moist convection and cloud formation, showing uncertainties in the impact of atmospheric warming profiles. By investigating the correlation between the warming profiles and CAO occurrence, the drivers behind the increasing prevalence of CAO clouds during winter and their decreasing prevalence during the shoulder seasons may be better understood.*

*Second, as indicated by CAOnet (Fig. 6a), CAO clouds are more frequently observed in regions of higher SSTs (Fig.A1b), revealing a potential SST dependence. However, as suggested by the CAO-SST correlation (Fig. A1a) and the relatively stable SSTs over the study period (Fig. A2),*

*SSTs are unlikely the driver of the observed CAO trends. However, in the Tropics, Sandu and Stevens, 2011 showed that closed-to-open cell transition occurred as a result of increasing SSTs. Consequently, as SSTs rise due to climate change, conditions may become more favorable for cumulus and open cells. This highlights the need for future research to investigate how changing SSTs affect the distribution of closed and open cell clouds, which is important for understanding the future radiative impact of CAOs.*

*Third, the impact of projected Arctic sea ice loss (DeRepentigny et al., 2016) could be studied using CAOnet. Observations indicate larger wintertime sea ice loss (Garcia-Soto et al., 2021), motivating future studies to conduct correlation analysis to assess how this loss has influenced the observed increase in wintertime CAO cloud cover. Investigating how total cloud cover and cloud properties evolve in relation to the location of the sea ice edge may help to quantify its effects on CAOs.*

To reflect this revamped section we have also updated the text in the conclusion on lines 525-527:

*While these regions align with relatively high SSTs, as well as frequent open cells to which CAOnet shows a high sensitivity, low CAO-SST correlation is found. Although this suggests SSTs to be of little importance for CAO trends, future studies may find clearer correlations as more data becomes available.*

**5)**

**Concerning the trends: The method that is being used has been called the 'median of pairwise slopes' method, and it may improve discussion of that approach if you used that name, since it is fairly intuitive.**

Thank you for the suggestion. This name was new to us, but we now see it is a common name for this method. We have now incorporated it into the description of our methods on lines 261-268:

*Utilizing the CAO cloud cover fraction of each region, trends were calculated using the median of pairwise slopes method, also known as the Theil-Sen trend estimator $T_{\mathrm{TS}}$ (Sen, 1968; Theil, 1950). This trend estimator is non-parametric, meaning it is independent of the distribution of the data, making it widely applied in climate data analysis (Collaud Coen et al., 2020; Gilbert, 1987; Yue et al., 2002). It is estimated using daily mean coverage of CAO clouds across the three regions, calculated as the median of all possible pairwise slopes:*

$$\mathrm{T_{TS}} = \mathrm{median}\left(\frac{y_i - y_j}{x_i - x_j}\right), \qquad (1)$$

*where $y_i$ denotes the cloud cover fraction on day $x_i$. Additionally, a confidence interval for this trend was estimated as the interval containing $\alpha$ (i.e. 95%) of the pairwise slopes, for which the median is represented as the trend in Eq. 1.*

**Further, it would really aid the work to do a little more to characterize the trends. A split between CAO frequency (how many days are CAOs detected, regardless of size), and amount when present (how much area is covered by CAO clouds) may help show whether CAO events are changing in frequency or size, or both.**

We agree that this would have been an excellent addition to our analysis and something that we initially set out to characterize. However, after testing multiple methods we found that the cloud cover trend presented in the manuscript was the most appropriate for our data and model. Below we describe the alternate methods we tried and some of the problems we ran into that ultimately led us to not using them.

First, we calculated trends in CAO cloud frequency on a grid point by grid point basis:

Almost no trends are detected when using the median of pairwise slopes (Theil-Sen) method and Mann-Kendall test for CAO frequency on a grid point by grid point basis. This is due to the fact that each grid point has values of either 1 or 0 depending on whether a CAO is present or not, and as most grid points have

zeros most of the time, most pairwise slopes will be 0, meaning that the final median will also be 0. One way to get around the dataset of ones and zeros is to take monthly means of each grid point, giving fractions instead. This will give us a trend estimate, but it reduces the number of time points from approximately 750 (25 years x 30 days) down to 25 (years), which is a strong limiting factor on how effectively we can detect the significance using the Mann-Kendall test. As shown in Figure A3, this gives some significant trends in the frequency of CAO clouds in the southern part of the domain in October and May, but not for the other months.

Although this does not disentangle whether the cloud cover trends are due to size or frequency, we have included this approach in the Appendix to further support our cloud coverage trend estimates. More specifically we have added Fig. A3 and the describing text to the Appendix on lines 552-569:

> In addition to the CAO cloud coverage trends presented in Fig. 8 and 9, monthly CAO cloud occurrence trends were calculated on a grid point by grid point basis, as shown in Fig. A3. This was accomplished using the median of pairwise slopes (Theil-Sen) method and Mann-Kendall test, as described in Sect. 2.3. However, as each grid point has values of either 1 or 0 depending on whether a CAO is present or not, and most grid points having zeros most of the time, most pairwise slopes will also be 0. Consequently, the final median and trend will also be 0. To overcome this limitation, monthly means for each grid point were calculated, resulting in monthly occurrence fractions that were used for the final trend estimate.

> Furthermore, a p-value was calculated using the Mann-Kendall test. A trend was deemed significant if it satisfied a false discovery rate of 0.2, following the Benjamini–Hochberg procedure (Benjamini & Hochberg, 1995). This method is recommended for addressing multiple hypothesis testing in atmospheric sciences (Wilks, 2016), allowing control over the expected fraction of false positives among the significant trends to be 20%.

> The resulting significant trends indicate a decreasing occurrence of CAO clouds in the southern region for both October and May over the past 25 years. This aligns with the cloud coverage trends for October shown in Fig. 8, and further supports the observed trend seasonality, suggesting a shortening of the CAO season.

[Figure]

Figure A3: Trends for each month calculated using the Theil-Sen estimator. Stippled points are insignificant by not satisfying a false discovery rate of 0.2.

To reference the new figure, the manuscript was also adapted on lines 427-429:

*Overall, the seasonality of the MBL instability and cloud cover trends, which is further supported by CAO cloud occurrence trends in Fig. A3, suggests clear seasonal drivers that may impact both the exchange of heat fluxes and the radiative balance of the Arctic.*

We also tried to characterize CAO cloud frequency trends for CAOs of different sizes:

First, we define CAO occurrence for any given day if a CAO clouds are present in more than x grid points. This will give us daily values of ones and zeros, corresponding to the presence of a CAO, which we can average over each month to calculate a frequency trend estimate. We tried thresholds greater than 0, as CAO clouds are nearly always present, and calculated trends for thresholds of 5, 15, 30, and 50 grid points to capture potential frequency trends in CAOs of varying sizes. However, these were not significant as seen in Tab. 1.

In total, we are unable to disentangle whether the cloud cover trends are due to size or frequency. We therefore chose to keep the focus on the cloud coverage trend estimates, with only the addition of Fig A3 and describing text in the appendix.

Table 1: Table showing trend estimates of CAO occurrence with a corresponding p-value. The thresholds indicate the required grid points to be covered by a CAO for the day to be considered a "CAO"-day.

| Month | Threshold 5 | Threshold 15 | Threshold 30 | Threshold 50 |
|-------|-------------|--------------|--------------|--------------|
| Sep | + 0.10, P=0.11 | + 0.02, P=0.47 | + 0.00, P=1.00 | + 0.05, P=0.44 |
| Oct | - 0.03, P=0.50 | - 0.04, P=0.32 | + 0.00, P=1.00 | + 0.00, P=1.00 |
| Nov | + 0.00, P=1.00 | - 0.05, P=0.56 | - 0.07, P=0.66 | - 0.11, P=0.47 |
| Dec | + 0.00, P=1.00 | + 0.01, P=0.66 | + 0.05, P=0.80 | + 0.09, P=0.37 |
| Jan | + 0.00, P=1.00 | + 0.00, P=1.00 | + 0.00, P=1.00 | + 0.00, P=1.00 |
| Feb | + 0.00, P=1.00 | + 0.00, P=1.00 | + 0.01, P=0.60 | - 0.04, P=0.73 |
| Mar | + 0.00, P=1.00 | + 0.00, P=1.00 | - 0.07, P=0.32 | + 0.00, P=1.00 |
| Apr | + 0.00, P=1.00 | - 0.10, P=0.13 | - 0.16, P=0.05 | - 0.15, P=0.15 |
| May | - 0.43, P=0.11 | - 0.27, P=0.11 | - 0.13, P=0.53 | - 0.13, P=0.38 |

**6)**

**Finally, it would strongly benefit the discussion of radiative characteristics to compare cloud amount observed when a CAO is occurring to cloud amount observed when it is not occurring. This is hinted at in the text, but is an essential result in order to actually add any value to a discussion of radiative impacts of CAOs.**

Thank you for the great suggestion. We have now added Fig. 7 and some discussion on lines 404-407 in the updated manuscript. While this figure does not directly show the radiative impact of CAOs, it shows that CAO clouds contribute up to 20 % of the cloud cover in the region. This highlights the importance of CAOs on the radiation balance and motivates the need to understand how CAO clouds are changing in a warming Arctic.

> *Despite the disagreements between the models, the clear seasonal pattern obtained suggests a corresponding pattern in terms of the cloud radiative impact. This is further illustrated in Fig. 7, showing that CAO clouds contribute up to 20% of the region's total cloud cover. With maximum coverage and cloud contribution occurring during winter and early spring, it is expected that potential trends during these months will be of the greatest significance for the radiative balance.*

Additionally, we have updated the manuscript on lines 455-461 to help motivate CAO's radiative impact.

> *Low-level clouds contribute the most to the Arctic surface radiative energy balance (Shupe & Intrieri, 2004), making trends in the low-level CAO clouds that contribute to up to 20 % of the total cloud cover in the study domain (Fig. 7) of high importance for Arctic warming. Their net radiative effect does however largely depend on solar radiation, which is negligible during winter for large portions of the study region. This results in a dominant longwave radiative warming from the increasing coverage of the CAO clouds during winter. With CAO presence accounting for up to 20 % of the region's clouds, they are likely responsible for a large longwave effect that may have contributed to an enhanced and region-dependent wintertime Arctic amplification (Rantanen et al., 2022).*

**1 Reviewer 2**

**1)**

**L23: Given it's the Arctic, perhaps say the seasonal radiative effect of CAO – they don't have a strong cooling effect in the winter, etc. This is a strong motivation for the study, so warrants more discussion of existing literature.**

[Figure]

Figure 7: Monthly CAO contribution to the total cloud cover in the study region following CAOnet.

Thank you for the suggestion. We agree that this is a strong point and have updated the manuscript on lines 25-32 to reflect this:

> As the Arctic has experienced significant warming in recent decades (Serreze & Barry, 2011), it is crucial to study how CAOs may be affected. This warming is especially pronounced during the winter months where CAOs are most prevalent (Dahlke et al., 2022; Fletcher et al., 2016), highlighting the potential of large impacts on CAOs in response to warming. Specifically, the strength of CAOs is projected to change (Landgren et al., 2019), which could affect cloud properties (Murray-Watson et al., 2023) and influence future warming. Furthermore, as the Arctic experiences polar night and day with highly seasonal variations in solar radiation, changes in CAO seasonality due to season-dependent warming may further impact the Arctic's radiative balance. This could potentially amplify or mitigate the significant warming observed in the region, underlining the importance of studying climatological shifts in the seasonality of CAOs.

**2)**

**L51: Can you explain more why the positive index is an issue? I understand the uncertainties in the analysis data, which is very good motivation. But, for example, studies which track the clouds (etc., Murray-Watson et al., 2023), they specifically identify clouds moving from ice to ocean with high indices, thus potentially limiting the false identification of the CAO clouds.**

Thank you for the suggestion. Yes, it is true that high indices will limit false detection, which is also addressed later on in the paper by suggesting a threshold to limit false detection. However, when using a high threshold, CAO clouds related to weaker instabilities and lower indices will be missed, potentially biasing further analysis to only be valid for the strongest CAO events. For a stronger motivation, we have addressed this by updating the manuscript on lines 49-62:

> By utilizing reanalysis products such as Modern-Era Reizing reanalysis products such as Mod Applications, Version 2 (MERRA-2, Gelaro et al., 2017), which provide global climate and weather data of the past at up to hourly resolution, earlier studies have defined a CAO as a model grid point where the index M is positive, indicating instability and the possibility for clouds (e.g. Fletcher et al., 2016; Murray-Watson & Gryspeerdt, 2024). This method provides an easy way to find the location of CAOs to use for further analysis, such as investigating cloud break-up. Despite the ease of use, reanalysis data introduces model biases especially in remote regions such as the Arctic with limited available observational data. In addition, uncertainties arise from the fact that a positive index does not necessarily result in the existence of a CAO cloud. While

*this can be addressed by requiring higher M indices (i.e. Murray-Watson et al., 2023), this may introduce biases from omitting clouds associated with weaker instabilities, skewing the analysis towards stronger CAO events. Furthermore, since the M index has been shown to decrease downwind (Murray-Watson et al., 2023), requiring higher M indices may result in the omission of clouds as they are advected downwind. Consequently, as there is no universally accepted M index for defining CAO clouds, this motivates the introduction of a phenomenological approach for defining CAOs that is based on the existence of clouds, and that is free of the biases introduced by modeling and reanalysis.*

**3)**

**Section 2.1.2: Variability in referencing: He et al is given as a reference for residual blocks improving performance, but nothing is given for the leaky ReLU. This might be convention for the ML community, though.**

Thank you for pointing this out. We have now added an appropriate reference (Maas et al., 2013) on line 140

**4)**

**Section 2.13: Why only five years of to train data if 25 were available? I thought usually the test/train split was 80:20, on a whole dataset? And are 500 swaths a robust evaluation dataset, if 15,200 swaths are available? Perhaps I am misunderstanding what warrants a statistically significant evaluation of ML techniques.**

Thank you for raising these questions. These are important questions, which we earlier have addressed but chose not to include in the paper. But to highlight the performance of the model with the training and split choices that we made, we have now included a description of the model uncertainties as described below.

Regarding the use of five years of train data:

Initially, when the performance of different model architectures were evaluated, only a smaller dataset containing 5 years of MODIS data was used to limit the computational costs. While we could have retrained the model on a larger dataset, we thought that this would not be worth the time and additional computational costs for a potential minimal performance increase.

Additionally, as mentioned in the paper on lines 164-175 in the updated manuscript, we checked that the classifications were not influenced by climatological shifts from it being trained on five of the last years of the study period. We include Fig. B1 to visualize this. Here we calculated the 95% confidence intervals from bootstrapping, showing insignificant trends in scores across all time periods. This suggests that the varying scores are a result of the random sampling when creating the evaluation label data.

In terms of the chosen 85:15 train-test split:

It is true that a 80:20 train-test split is a good rule of thumb, but any split may be chosen depending on for instance data size and model architecture. However in general, the idea with any chosen split is to balance the variance of the model estimates with the variance of the performance metric. Following the central limit theorem, the test loss standard error behaves as $\sigma/\sqrt{n_{\text{samples}}}$, where $\sigma$ is the standard deviation of the per-sample loss. In our case, we have 15,200 swaths split into approximately 600,000 patches, translating to $n_{\text{samples}} \approx 90,000$ patches, giving a very small standard error for our loss. This supports using a relatively small test-split while keeping most data for training.

Additionally, in Fig. B2 we see that the train and test loss closely follows each other without large fluctuations from epoch to epoch during training. This is consistent with a low test loss standard error and no overfitting. Consequently, there seem to be no implications of our chosen train-test split.

Our reasoning for a 500-image evaluation dataset:

Indeed, with a total of 15,200 swaths available (for approximately each 5-year period), one could argue that we should have used more than 500 swaths (from a 25-year period) to accurately evaluate model performance. While a doubling of the amount of swaths would scale the standard error of our metrics with $1/\sqrt{2}$, giving us smaller uncertainties, we thought that the added benefit was not worth it relative to the additional labeling time and effort. To see why, we can calculate the uncertainties of our evaluation

[Figure]

Figure B1: Score bias for the different 5-year periods. Uncertainty bars indicate the bootstrapped 95% confidence interval.

[Figure]

Figure B2: Autoencoder training history showing the train and test loss at each iteration through the complete training dataset.

[Figure]

Figure 3: Confusion matrices for CAOnet (a), MERRA-2 with $M > 3.75\,\mathrm{K}$ (b), and MERRA-2 with $M > 0\,\mathrm{K}$ (c). The y-axis represents predicted classes by the models, while the x-axis represents the labeled classes. Positive corresponds to classified CAO, while negative means no CAO was classified. For each colored box, the lower number corresponds to number of classified patches, while the upper percentage corresponds to the rate of that predicted class relative to the actual class. The lower gray row shows the recall and true negative rate (upper) and number of patches labeled as that actual class (lower). The rightmost gray column shows the precision for positive and negative prediction (upper) and number of patches corresponding to that predicted class (lower). Finally the lower right corner in dark gray shows the total accuracy (upper) and total number of patches classified (lower).

metrics by performing bootstrap, which gives standard errors ranging from 0.5-1.7%. To show these low uncertainties, we have chosen to add bootstrap to the methods and show the standard errors in the results.

On lines 183-186 in the methods, we have added:

> *To quantify uncertainty due to the relatively small evaluation dataset, swath-level bootstrap resampling was performed. From the original 500-swath evaluation dataset, 10,000 new 500-swath bootstrapped replicates were sampled with replacement. For each replicate, score metrics were calculated, before the standard deviation of the 10,000 bootstrap metric values was computed to provide a bootstrap estimate of the standard error.*

On lines 287-293 in the results, we have added:

> *This combination achieves an accuracy and corresponding bootstrapped standard error of 85.4±0.5%, a recall of 72.4±1.3%, a true negative rate of 90.4±0.6% and precision of 74.6±1.3%, as can be seen in Fig. 3a. Meanwhile when evaluating previously used CAO criteria for the $M$ index, it was found that a threshold of $M > 3.75\,K$ ($M_{3.75}$) performed the best, in contrast to the commonly used threshold of $M > 0\,K$ ($M_0$) (e.g. Fletcher et al., 2016; Murray-Watson & Gryspeerdt, 2024). With $M_{3.75}$, an accuracy of 78±0.7%, a recall of 71.1±1.7%, true negative rate of 80.7±1.1% and precision of 59.0±1.2% is reached (see Fig. 3b), which is better than when using a threshold of $M_0$ reaching an accuracy of 67.4±0.9%, a recall of 93.5±0.8%, true negative rate of 57.3±1.5% and precision of 46.0±1.2% (see Fig. 3c).*

**5)**

**Line235: Are they weaker CAOs, or CAOs further along in their development?**

Thank you for pointing this out. They may just as well be further along in their development, giving lower $M$ indices. We have made some updates on lines 255-259 to reflect this:

*The northern region was chosen based on typical CAO trajectories, capturing CAOs earlier on in their development due to the proximity to the sea ice edge. This typically results in larger concentrations of closed cells. In contrast, the southern region was chosen for its distance from the sea ice edge, capturing CAOs later on in their development (Murray-Watson et al., 2023), where a greater prevalence of open cells are expected (Brümmer, 1999).*

**6)**

**Lin262: Is there any way to rationalise why there are two CAO clusters? Is it early/late in development? What characteristics split them?**

Thank you for the comment. Indeed, we were initially hoping that additional information could be extracted from the two the clusters (e.g. open vs cloud cells). But we found that there was no real discernible or systematic difference between the two clusters.

**7)**

**Section3.1: How do you account for overlying clouds affecting the M-based classification in the imagery, which you're automatically filtering out of the CAOnet one? Is it a fair comparison?**

This is a great point and definitely something that should be clarified. We do not filter out the high clouds when evaluating the $M$ index, which potentially leads to an unfair comparison when CAO clouds are obscured by high clouds. However, it is hard to evaluate CAO cloud presence during these conditions, limiting our ability to validate the M-index. To clarify this limitation, we have adapted the manuscript on lines 239-241:

*It is also important to note that in situations where overlying clouds were present, it was impossible to determine the presence of CAO clouds below. While this limitation skewed the M index scores towards scenarios without overlying clouds, it optimized the final M index threshold for conditions where CAO clouds are especially relevant for cloud radiative properties.*

**8)**

**Line275: The high M index probably doesn't capture the open cells well because they are downstream of the CAO development, no? So are they not still CAO clouds, just advected downstream? Tracking studies seem to show that M decreases downwind of the sea ice edge, so using this gridbox-by-gridbox metric with M 3.75 will necessarily miss these clouds. This may also affect later discussion about Climatology, etc.**

Thank you for your comment. You point out important aspects, which highlight why the $M$ index is suboptimal when studying these clouds, even when optimizing the $M$ threshold. We have now included these aspects on lines 55-62 as a motivation for the use of CAOnet when studying the development of CAO clouds.

*In addition, uncertainties arise from the fact that a positive index does not necessarily result in the existence of a CAO cloud. While this can be addressed by requiring higher M indices (i.e. Murray-Watson et al., 2023), this may introduce biases from omitting clouds associated with weaker instabilities, skewing the analysis towards stronger CAO events. Furthermore, since the M index has been shown to decrease downwind (Murray-Watson et al., 2023), requiring higher M indices may result in the omission of clouds as they are advected downwind. Consequently, as there is no universally accepted M index for defining CAO clouds, this motivates the introduction of a phenomenological approach for defining CAOs that is based on the existence of clouds, and that is free of the biases introduced by modeling and reanalysis.*

This is in addition to some of the disadvantages of the M index, which are already described in the paper:
On lines 379-386, we suggest that one should optimally vary the index threshold as well as implementing a varying pressure level for calculating M as the clouds are advected. Additionally, we suggest that the M

index threshold should optimally be varied based on season on lines 398-403. This is to be able to capture more of the weaker CAOs that occur in the shoulder seasons.

While we could define a CAO from the M index just off the sea ice and follow the air as it is advected, it is impossible to know when the clouds eventually dissipate. In total, this shows that quite extensive research on optimal thresholds is required when one wishes to study CAO clouds. Consequently, using a single threshold of 3.75 K for the whole study region becomes an easy-to-implement threshold to roughly predict the presence of a CAO cloud. However, as this approach is by no means perfect, it underscores the advantages of using CAOnet when studying these clouds.

**9)**

**3.3: I'm not sure some of the figures mentioned here align with what Figure 7 shows. Also, broadly, I think this section could do with some more mechanistic analysis. They discuss the literature, but it would be helpful if they could show how their tools could be used to answer these questions. It is more speculative than explanatory.**

Thank you for also pointing out the misaligning Figure and text. The figure was updated last minute with one additional year of data, and some of the text had not been updated accordingly. The addition slightly altered the p-values for March and December, resulting in the December trend becoming significant and the March trend becoming insignificant at a 0.05 level. The text is now updated by replacing "March" with "December" on line 416 and line 418 to reflect the new figure.

Additionally, we have toned down a lot of the speculation and rather focused on motivating future research by providing potential important factors. We updated Sec. 3.3.2 on lines 431-453 to:

[revised manuscript text omitted]

Finally, we removed some speculation around the influence of warming SST on the trends in the southern region in section 3.3.4.

**10)**

**L525: Similarly, the conclusions state that the trends are driven by faster warming of the air compared to the sea, but this isn't necessarily shown.**

Thank you for the comment. Now, after changing the trend attribution section we have also adjusted the conclusions to rather motivate future research to look into trends in warming profiles and SST. Additionally, we chose to add a trend to Fig. A2 , showing the relatively low SST trends in comparison to atmospheric warming. The conclusion section on lines 514-543 is now updated to:

*Clouds associated with CAOs are important for the Arctic radiative energy budget, particularly in the regions surrounding the Norwegian sea, Barents sea and Northern Atlantic due to their extensive coverage. In the rapidly warming Arctic, it becomes important to understand how these clouds respond to climate change, as changes in their prevalence and properties may either amplify or dampen local and global warming. To explore shifts in CAO cloud coverage, a phenomenological CAO cloud classification method named CAOnet has been developed. This approach, based on MODIS data and machine learning, has demonstrated promising results compared to the traditional marine cold air outbreak index (M) in detecting CAO clouds. Additionally, an M index threshold of 3.75 K has been identified for optimal detection of CAO clouds, providing future studies a basis for the instability required for CAO cloud formation.*

*By employing CAOnet, a CAO climatology focusing on clouds rather than MBL instability has been produced. In contrast to the M index calculated using MERRA-2 reanalysis, this climatology has provided an alternative perspective on CAOs, highlighting the Norwegian coast and the North Atlantic region south of Iceland as key areas for CAO clouds and their associated radiative impact. While these regions align with relatively high SSTs, as well as frequent open cells to which CAOnet shows a high sensitivity, low CAO-SST correlation is found. Although this suggests SSTs to be of little importance for CAO trends, future studies may find clearer correlations as more data becomes available.*

*Utilizing data from the past 25 years, CAO clouds have been found to contribute to up to 20 % of the total cloud cover in the study region. This underscores the importance of the observed trends, revealing a shortening of the CAO season as indicated by a 10 % increase in wintertime CAO cloud cover and nearly a 20 % decrease during the shoulder months of October, November and April. These shifts are in large part linked to changes in open cell cloud cover as well as changes in the MBL (as indicated by the M index). By utilizing CAOnet, future studies have an easy-to-use database to investigate the potential drivers of these trends, such as correlation analysis with atmospheric warming profiles and the position of the sea ice edge.*

*Due to the lack of incoming solar radiation during winter, the radiative impact of the observed wintertime cloud cover trends are likely dominated by the terrestrial radiative warming effect. This may have contributed to the anomalously strong and region-dependent wintertime Arctic amplification. Conversely, during the shoulder months when decreasing cloud cover is observed, the increased solar radiative effect introduces uncertainties regarding the overall radiative effect of the cloud cover trends.*

*While these trends do not directly indicate future changes in the Arctic's radiative energy balance, they clearly indicate that CAO clouds are influenced by climate change. This emphasizes the importance of accurately characterizing the radiative impact of these clouds and understanding their role in the local and global climate system. This motivates future work to utilize CAOnet together with spaceborne flux products such as from CERES, to accurately uncover the radiative impact of the observed CAO trends.*

**11)**

**Similarly, considering the radiative impact of CAOs in a changing Arctic was a strong motivation for the study, some analysis would be appreciated. Conclusions are inferred from changes in trends etc., but again, this is more speculative.**

Thank you for the comment. We agree and have with the changes based on the previous comments, updated the trend discussion and conclusions. This includes revamping the entire radiative impact section to motivate future studies to use CAOnet together with other radiative products. Additionally, by adding Fig. 7 which shows that CAOs contribute up to 20 % of the region's total cloud cover, we further motivate their importance. This has transformed the entire Trends section (3.3) to become less speculative, shifting the radiative focus from a speculative impact discussion to one that underscores CAO's potential impact and motivates future studies to investigate this using CAOnet.